# NEURAL FLOW SAMPLERS WITH SHORTCUT MODELS

## ABSTRACT

Sampling from unnormalized densities presents a fundamental challenge with wide-ranging applications, from posterior inference to molecular dynamics simulations. Continuous flow-based neural samplers offer a promising approach, learning a velocity field that satisfies key principles of marginal density evolution (e.g., the continuity equation) to generate samples. However, this learning procedure requires accurate estimation of intractable terms linked to the computationally challenging partition function, for which existing estimators often suffer from high variance or low accuracy. To overcome this, we introduce an improved estimator for these challenging quantities, employing a velocity-driven Sequential Monte Carlo method enhanced with control variates. Furthermore, we introduce a shortcut consistency model to boost the runtime efficiency of the flow-based neural sampler by minimizing its required sampling steps. Our proposed Neural Flow Shortcut Sampler empirically outperforms existing flow-based neural samplers on both synthetic datasets and complex n-body system targets.

## 1 INTRODUCTION

We consider the task of sampling from unnormalised densities $\pi(x) = \frac{\rho(x)}{Z}$ for a $d$-dimensional variable $x \in \mathbb{R}^d$, where $Z := \int \rho(x)\,\mathrm{d}x$ denotes the unknown partition function. This task is fundamental in probabilistic modeling and scientific simulations, with applications in Bayesian inference (Neal, 1993), drug discovery (Xie et al., 2021), and material design (Komanduri et al., 2000). However, achieving efficient sampling remains challenging, especially when dealing with high-dimensional and multi-modal distributions. Established methods, primarily Markov Chain Monte Carlo (MCMC), often require long convergence times and extended simulations to obtain uncorrelated samples (Brooks et al., 2011).

Neural samplers have emerged as an alternative class of methods, which leverage recent developments in generative models, such as normalising flows (Midgley et al., 2023) and latent variable models (He et al., 2024). Very recent work in this line also took inspirations from diffusion models and flow matching (Ho et al., 2020; Lipman et al., 2022). For instance, Sadegh et al. (2024) introduced a diffusion-based sampler that trains a score network via Monte Carlo estimation. Meanwhile, continuous-time flow-based approaches (Máté & Fleuret, 2023; Tian et al., 2024; Albergo & Vanden-Eijnden, 2025) aim to learn a continuous transformation from a simple distribution to the target density, by designing residual loss functions derived from continuous dynamics (e.g., the continuity equation (Villani et al., 2009)). Related works also employ an optimal control perspective to design the learning objective (Zhang & Chen, 2022). These advanced approaches show promising empirical performance, often comparable to results obtained via MCMC.

Still current diffusion and flow-based neural samplers face challenges. Specifically, continuous flow-based approaches often require estimating or learning the evolving partition functions (or their derivatives) along the transformation path, which suffers from high variance via importance sampling (Tian et al., 2024) or low accuracy when estimated via gradient-based optimisation (Máté & Fleuret, 2023; Albergo & Vanden-Eijnden, 2025). Meanwhile, diffusion and flow-based neural samplers rely on simulating the learned differential equations with accurate numerical solvers, posing a challenge in dynamically adjusting their runtime budget without significant impact on sample quality.

In this work, we propose Neural Flow Shortcut Sampler (NFS$^2$), which is flexible, simple, and able to dynamically adjust its sampling budget with marginal impact on sample quality. Our approach

trains the flow-based sampler by minimising a residual loss derived from the continuity equation, and we tackle the challenges using strategies summarised below:

- We propose a low-variance bootstrap estimator for the partition function time derivative using Sequential Monte Carlo (Gordon et al., 1993; Liu & Chen, 1998). Our approach features a velocity-driven transition kernel as proposal, and a Stein identity-based (Stein, 1981) control variate for variance reduction.
- We extend the shortcut consistency method (Frans et al., 2024) proposed for generative models to flow-based samplers. Our approach returns a dynamically adjustable sampler that can simulate samples within arbitrary computational budget without significant compromise on sample quality.

Across challenging experiments-sampling from synthetically constructed multi-modal targets such as a 40-mode mixture of Gaussians and the 32-dimensional Many Well (Midgley et al., 2023), alongside the classic Lennard-Jones interatomic potential in a 13-particle system (Jones, 1924) and Double Well potential in a 4-particle system-NFS[2] consistently demonstrates strong performance competitive with leading contemporary neural samplers. These results underscore the efficacy of our method.

## 2 BACKGROUND

**Continuous Normalising Flows.** Continuous Normalising Flows (CNFs) (Grathwohl et al., 2018) are generative models that learn a continuous, invertible transformation from a simple base distribution $p_0$ to a complex target distribution $p_1$. This transformation is defined by the solution to an ordinary differential equation (ODE) (1) that describes the trajectory $x_t$ of a sample starting from $x_0 \sim p_0$. The density $p_t(x_t)$ of the transported samples $x_t$ can be calculated by integrating the instantaneous change of variables (Grathwohl et al., 2018) from $t = 0$ to $t = 1$ (2):

$$\frac{dx_t}{dt} = v_t(x_t), \quad \text{for } t \in [0, 1], \tag{1}$$

$$\partial_t[\log p_t(x_t)] = -\nabla_{x_t} \cdot v_t(x_t) \implies \log p_1(x_1) = \log p_0(x_0) - \int_0^1 \nabla_{x_s} \cdot v_s(x_s) ds, \tag{2}$$

where $v_t : \mathbb{R}^d \to \mathbb{R}^d$ is a time-dependent velocity field, typically parameterised by a neural network.

**The Continuity Equation.** The underlying equation governing the evolution of the probability density $p_t(x)$ at a fixed point $x$ under the influence of the velocity field $v_t(x)$ is the continuity equation (3) (Villani et al., 2009). It states that if the velocity $v_t$ generates the probability path $\{p_t(x)\}_{t \in [0,1]}$ from an initial distribution $p_0$ to a target $p_1$. This equation ensures the conservation of probability mass. Notably, a connection between the continuity equation and CNFs is revealed by considering the total derivative of $\log p_t(x_t)$ w.r.t. $t$:

$$\forall x \in \mathbb{R}^d, \quad \partial_t p_t(x) = -\nabla_x \cdot [p_t(x)v_t(x)] \tag{3}$$

$$\partial_t \log p_t(x_t) + \nabla_{x_t} \log p_t(x_t) \cdot v_t(x_t) = -\nabla_{x_t} \cdot v_t(x_t) \tag{4}$$

## 3 RELATED WORK

Sampling from probability distributions remains a key research challenge. Classical Monte Carlo methods, such as Annealed Importance Sampling (Neal, 2001) and Sequential Monte Carlo (Gordon et al., 1993; Liu & Chen, 1998; Del Moral et al., 2006), are benchmarks but often computationally costly and slow to converge (Roberts & Rosenthal, 2001). Amortized variational methods like normalizing flows (Rezende & Mohamed, 2015) and latent variable models (He et al., 2024) provide alternatives for approximating target distributions. Hybrid approaches (Wu et al., 2020; Zhang et al., 2021; Geffner & Domke, 2021; Matthews et al., 2022; Midgley et al., 2023) combine MCMC and variational inference, showing promise by leveraging the strengths of both.

Advancements in generative modeling have applied diffusion models (Ho et al., 2020) and flow matching (Lipman et al., 2022) to sampling. Vargas et al. (2023); Nusken et al. (2024), for example, use diffusion processes for sample learning, requiring SDE simulation for training. The quest for

simulation-free training led to methods like iDEM (Sadegh et al., 2024) and BNEM (OuYang et al., 2024). iDEM uses a bi-level scheme, iteratively generating samples and score matching with Monte Carlo (MC) estimates. BNEM targets MC-estimated noised energies, which reduces variance. Woo & Ahn (2024) offer a similar variant targeting MC-estimated vector fields in a flow matching framework.

PINN-based approaches using the continuity equation (Tian et al., 2024; Máté & Fleuret, 2023; Albergo & Vanden-Eijnden, 2025) also offer simulation-free training objectives. For instance, NETS (Albergo & Vanden-Eijnden, 2025) adds a learned drift to AIS to reduce variance, and Máté & Fleuret (2023) proposes a learnable transport path interpolation; both learn $\partial_t \log Z_t$ as part of their training. Alternatively, LFIS (Tian et al., 2024) uses a similar loss with an importance-sampled estimator for $\partial_t \log Z_t$, risking high variance.

Stochastic optimal control (SOC) (Pavon, 1989; Tzen & Raginsky, 2019) are also applied to sampling. Zhang & Chen (2021), for example, introduced Path Integral Sampler (PIS) based on sampling-optimal control links (Chen et al., 2016). Berner et al. (2022) further connected optimal control with SDE-based generative modeling (Kloeden et al., 1992). Recently, Adjoint Sampling (Havens et al., 2025) used this principle in an on-policy method for better efficiency and scalability.

## 4 NEURAL FLOW SHORTCUT SAMPLER

To sample from the unnormalised target density $\pi \propto \rho$, we learn a time-dependent velocity field $v_t(x; \theta)$, parametrised by $\theta$, that transforms samples from a simple base distribution $p_0$ into samples from the target $p_1 := \pi$. Our approach employs Eq. (4) to construct a loss for learning $v_t(x; \theta)$, followed by the further improvement of using shortcut models.

### 4.1 LEARNING FLOW SAMPLERS VIA A PINN LOSS

We start from constructing a probability path $\{p_t(x)\}_{t \in [0,1]}$ using an annealing interpolation (Brooks et al., 2011):

$$p_t(x) = \frac{\tilde{p}_t(x)}{Z_t}, \quad \tilde{p}_t(x) \triangleq p_1^t(x) p_0^{1-t}(x), \quad Z_t = \int \tilde{p}_t(x) dx. \tag{5}$$

We wish to learn a velocity field $v_t(x; \theta)$ which, together with $\{p_t(x)\}_{t \in [0,1]}$ defined by Eq. (5), satisfies the continuity equation Eq. (3). This inspires a PINN objective (Máté & Fleuret, 2023; Albergo & Vanden-Eijnden, 2025) designed as the point-wise residual error when plugging-in $v_t(x; \theta)$ and $p_t(x)$ into Eq. (4):

$$\mathcal{L}(\theta) = \mathbb{E}_{q_t(x), w(t)}[\delta_t^2(x; v_t(\cdot; \theta))], \delta_t(x; v_t) \triangleq \partial_t \log p_t(x) + v_t(x) \cdot \nabla_x \log p_t(x) + \nabla_x \cdot v_t(x) \tag{6}$$

Here $w(t)$ denotes the time-weighting distribution for $t \in [0, 1]$, and $q_t(x)$ represents the source of samples for evaluating the loss at time $t$. In principle, $q_t(x)$ is only required to share the same support as $p_t(x)$ (and thus the target $\pi$) so that Eq. (4) holds for all $x \in \text{supp}(p_1)$ when $\mathcal{L}(\theta) = 0$. In practice, using data points from a simple $q_t(x)$ distribution such as a uniform distribution over $\mathbb{R}^d$ is sub-optimal, necessitating a sophisticated strategy for generating samples $x$ that are used to evaluate the point-wise residual loss.

### 4.2 IMPROVED ESTIMATES OF $\partial_t \log Z_t$

Computing the time derivative term $\partial_t \log p_t(x)$ in the loss objective Eq. (6) introduces additional complexity, due to the time derivative of the log partition function:

$$\partial_t \log p_t(x) = \partial_t \log \tilde{p}_t(x) - \partial_t \log Z_t, \ \partial_t \log Z_t = \partial_t \log \int \tilde{p}_t(x) \, dx = \mathbb{E}_{p_t(x)}[\partial_t \log \tilde{p}_t(x)]. \tag{7}$$

Albergo & Vanden-Eijnden (2025); Máté & Fleuret (2023) propose to learn $\partial_t \log Z_t$ jointly together with the velocity field $v_t(x; \theta)$ via gradient descent, which is sub-optimal in more complex scenarios, as demonstrated in our experimental results. On the other hand, Tian et al. (2024) estimate this time derivative using importance sampling (see Appendix A.1 for details):

$$\partial_t \log Z_t \approx \sum_{k=1}^{K} \bar{w}_t^{(k)} \partial_t \log \tilde{p}_t(x_t^{(k)}), \quad x_t^{(k)} \sim q_t(x), \ w_t^{(k)} = \frac{p_t(x_t^{(k)})}{q_t(x_t^{(k)})}, \ \bar{w}_t^{(k)} = \frac{w_t^{(k)}}{\sum_{j=1}^{K} w_t^{(j)}}. \tag{8}$$

---

**Algorithm 1** Velocity-driven transition kernel from time-steps $t_{m-1}$ to $t_m$

---

**Require:** $\{x_{t_{m-1}}^{(k)}\}_{k=1}^{K}, \{w_{t_{m-1}}^{(k)}\}_{k=1}^{K}, v_{t_{m-1}}(\cdot; \theta), \tilde{p}_{t_m}, \tilde{p}_{t_{m-1}}, \gamma$

1: Velocity move proposal: $\tilde{x}_{t_m}^{(k)} \leftarrow \hat{x}_{t_{m-1}}^{(k)} + \gamma v_{t_{m-1}}(\hat{x}_{t_{m-1}}^{(k)}; \theta)\Delta t$

2: MCMC refinement: $x_{t_m}^{(k)} \sim \text{MCMC}(\cdot | \tilde{x}_{t_m}^{(k)})$ targeting $p_{t_m}$            ▷ e.g., HMC

3: Update importance weights: $w_{t_m}^{(k)} \leftarrow w_{t_{m-1}}^{(k)} \frac{\tilde{p}_{t_m}(x_{t_m}^{(k)})}{\tilde{p}_{t_{m-1}}(x_{t_m}^{(k)})}$

4: **return** $\{x_{t_m}^{(k)}\}_{k=1}^{K}, \{w_{t_m}^{(k)}\}_{k=1}^{K}$

---

However, importance sampling can suffer from high variance if the proposal distribution differs significantly from the target $p_t(x)$. To address this issue, we propose a strategy employing Sequential Monte Carlo (SMC) (Gordon et al., 1993; Liu & Chen, 1998) estimation in conjunction with a velocity-driven Hamiltonian Monte Carlo (HMC) kernel (Duane et al., 1987).

**Velocity-driven SMC.** Concretely, consider discrete-time steps $0 = t_0 < \cdots < t_M = 1$, the key ingredients of SMC are proposals $\{\mathcal{F}_{t_m}(x_{t_{m+1}} | x_{t_m})\}_{m=0}^{M-1}$ and weighting functions $\{w_{t_m}\}_{m=0}^{M}$. The choice of proposals is critical: a poor proposal can lead to particle degeneracy; the proposal mechanism must be efficient, as the generated particles are used to define $q_t(x)$ for evaluating the residual loss. In light of these considerations, we propose incorporating both the velocity model $v_t(x; \theta)$ and additional HMC refinement steps as the transition kernel in the SMC framework (Van Der Merwe et al., 2000). In a nutshell, the velocity-driven SMC algorithm runs by first drawing $K$ particles of $x_{t=0}^{(k)} \sim p_{t=0}$ and setting $w_{t=0}^{(k)} = \frac{1}{K}$, then sequentially repeating sampling steps in Alg. 1 for $t = t_{1:M}$ until reaching $t_M = 1$. Within a transition, the ESS $= 1/\sum_{k=1}^{K}(\tilde{w}_{t_m}^{(k)})^2$ is tracked and if it drops below a predefined threshold, a systematic resampling step is performed to draw $K$ new particles from the current set of weighted particles, and their weights are reset to $1/K$. The velocity-driven MCMC kernel operates by using the velocity model $v_t(x; \theta)$ to provide an informed initialization for the HMC steps (See Alg. 1). As training progresses, the velocity model becomes increasingly better in generating the distribution path $\{p_t(x)\}_{t \in [0,1]}$, which reduces the needed correction steps by HMC. The proposal also benefits from MCMC's stochastic exploration to get a diverse set of samples from the SMC procedure and the unbiasedness of the SMC procedure (Del Moral, 2004).

To demonstrate the efficacy of the proposed velocity-driven SMC sampler, we measure the Mean Squared Error (MSE) of the $\partial_t \log Z_t$ estimation against the reference (obtained from very long MCMC chains) for varying numbers of MCMC steps within each SMC transition (see Figure 1) in Lennard-Jones system. For a reasonably well-trained velocity field, this strategy of improved initialisation demonstrably reduces estimation error. This synergy between the velocity-informed proposal and MCMC refinement can accelerate convergence and help preserve particle diversity, enhancing robustness in complex settings.

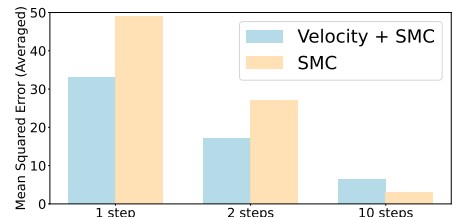

Figure 1: MSE of the estimated $\partial_t \log Z_t$.

**Variance Reduction via Stein Control Variates.** To further reduce the variance of our estimation, a key observation is that for any velocity $v_t$, the following identity holds

$$\partial_t \log Z_t = \underset{c_t}{\arg\min} \, \mathbb{E}_{p_t}(\xi_t(x; v_t) - c_t)^2, \quad \xi_t(x; v_t) \triangleq \partial_t \log \tilde{p}_t(x) + \nabla_x \cdot v_t(x) + v_t(x) \cdot \nabla_x \log p_t(x). \quad (9)$$

See Appendix B.1 for proof. Thus, one can calculate the optimal $c_t$ via $\partial_t \log Z_t = \mathbb{E}_{p_t}[\xi_t(x; v_t)]$, which can be approximated via Monte Carlo estimation via e.g., the proposed velocity-driven SMC procedure. In other words, the time derivative can be estimated as $\partial_t \log Z_t \approx \sum_{k=1}^{K} \bar{w}_t^{(k)} \xi_t(x_t^{(k)}; v_t)$, using the normalized weights $\bar{w}_t^{(k)}$ and particles $x_t^{(k)}$ from the SMC procedure with proposal and transition steps defined as in Alg. 1.

Empirically, we observe that Eq. (9) achieves lower variance compared to a simple Monte Carlo estimator $\partial_t \log Z_t \mathbb{E}_{p_t} \partial_t \log \tilde{p}_t(x) \approx \sum_{k=1}^K \bar{w}_t^{(k)} \log \tilde{p}_t(x_t^{(k)})$, and it sometimes leads to better optimisation of the training objective Eq. (6). We visualize the comparison of the standard deviation of the two estimation methods in Figure 2, with the corresponding loss plots deferred to Figure 11. This variance reduction effect can be explained from a control variate perspective (detailed in Appendix B.2). Specifically, the additional term $\nabla_x \cdot v_t(x) + v_t(x) \cdot \nabla_x \log p_t(x)$ is the Stein operator over a velocity field $v_t(x)$ under distribution $p_t(x)$, and with mild assumptions, the following Stein's identity holds (Stein, 1981):

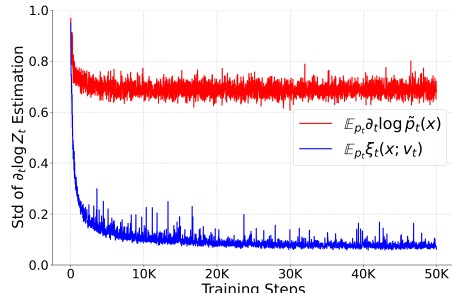

Figure 2: Variance reduction as shown by the standard deviations of the estimators.

$$\mathbb{E}_{p_t(x)}[\nabla_x \cdot v_t(x) + v_t(x) \cdot \nabla_x \log p_t(x)] = 0 \quad \Rightarrow \quad \mathbb{E}_{p_t(x)}[\xi_t(x; v_t)] = \partial_t \log Z_t. \tag{10}$$

Therefore, this Stein operator term acts as a zero-mean control variate (Liu et al., 2017) for estimating $\partial_t \log Z_t$. By estimating $\mathbb{E}_{p_t}[\xi_t(x; v_t)]$, we are implicitly using a control variate derived from the velocity field $v_t$ and the score function $\nabla_x \log p_t(x)$.

### 4.3 SHORTCUT MODEL FOR FURTHER ACCELERATIONS

Generating samples from a learned velocity model $v_t(x; \theta)$ typically requires simulating the underlying ODE $dx/dt = v_t(x; \theta)$ with numerical solvers (e.g., the Euler-Maruyama scheme with small time steps $\Delta t$). This can lead to a high number of neural network function evaluations and thus a significant computational cost during sampling. To address this, we draw inspiration from recent shortcut models (Frans et al., 2024) and introduce an additional consistency regularisation term.

The core idea is to parametrise a shortcut model $s_t(x_t, d; \theta)$ that directly predicts the *average* velocity required to traverse a finite time interval of duration $d$. The position at $t + d$ is then approximated by $x_{t+d} \approx x_t + s_t(x_t, d; \theta)d$. The instantaneous velocity $v_t(x; \theta)$ from previous sections is now considered as the $d \to 0$ limit of this shortcut model, i.e., $v_t(x; \theta) = s_t(x, 0; \theta)$.

To ensure that these shortcut predictions are consistent across different interval lengths, we enforce a regularisation based on trajectory consistency. Specifically, taking a single large step over an interval $d$ should yield a similar average velocity to taking two consecutive smaller steps that span the same total interval. Moreover, we propose to generalise the consistency condition in Frans et al. (2024) by choosing a random fraction $\alpha \in [0, 1]$ which splits the total interval $d$ into two parts: a first step of duration $\alpha d$ and a second step of duration $(1 - \alpha)d$. The state after the first sub-step is estimated as $x_{t+\alpha d} = x_t + s_t(x_t, \alpha d; \theta)\alpha d$. Then the target average velocity over the full interval $d$ is constructed from the two sub-steps, where the two terms represent the velocity for the two parts respectively:

$$s_{\text{target}}(x_t, t, d, \alpha; \theta) = \alpha s_t(x_t, \alpha d; \theta) + (1 - \alpha)s_{t+\alpha d}(x_{t+\alpha d}, (1 - \alpha)d; \theta).$$

The final objective combines the residual loss Eq. (6) (using the $d = 0$ limit, $s_t(\cdot, 0; \theta)$) with a consistency loss, averaged over sampled intervals $d$ and random splits $\alpha$:

$$\mathcal{L}(\theta) = \mathbb{E}_{q_t(x), w(t)} \left[ \delta_t^2(x; s_t(\cdot, 0; \theta)) + \lambda \mathbb{E}_{p(d), p(\alpha)} \| s_t(x, d; \theta) - s_{\text{target}}(x, t, d, \alpha; \theta_{\text{sg}}) \|_2^2 \right]. \tag{11}$$

Here $\lambda$ is a weighting coefficient, $p(d)$ is a distribution over interval lengths (e.g., uniform over $\mathcal{U}(0, 1)$), $p(\alpha)$ is typically uniform $\mathcal{U}(0, 1)$, and $\theta_{\text{sg}}$ indicates parameters with stopped gradients within $s_{\text{target}}$. We refer to the proposed methods as *neural flow shortcut samplers* (NFS$^2$), using 128 sampling steps by default unless specified otherwise. The training and sampling procedures are summarised in Algorithms 3 and 4 in Appendix C, respectively.

### 4.4 ADDITIONAL DESIGN CHOICES: REFERENCE DISTRIBUTION $q_t(x)$ AND NETWORK ARCHITECTURES

**Further augmentation for $q_t(x)$.** A crucial aspect of the training is the selection of the distribution $q_t(x)$. An effective $q_t(x)$ should concentrate computational effort on regions relevant to the evolving probability path $p_t(x)$. This challenge has also been observed in various PINN-based methods, requiring more sophisticated strategies (Wu et al., 2023). We propose that the distribution $q_t(x)$ be

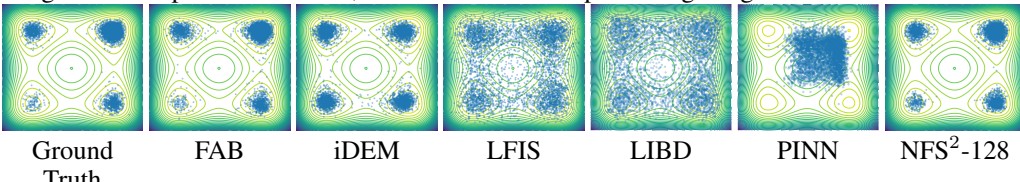

Figure 3: Samples of GMM-40, with contour lines representing the ground truth distribution.

Figure 4: 2D marginal samples from the 1st and 3rd dimensions of MW-32.

defined by data points generated as follows: approximate samples of $p_t(x)$ are first obtained during the SMC sampling process; these samples are then perturbed and undergo an additional residual-based resampling stage as described in Appendix D. We found these steps critical as alternative strategies failed to adequately learn a good sampler.

**Network architectures for shortcut models**  We employ neural network architectures for the shortcut model $s_\theta(x, t, d)$ tailored to the specific characteristics of each experimental task. For simpler, lower-dimensional tasks, we use Multi-Layer Perceptrons (MLPs), while for the more complex particle system (e.g., Lennard-Jones potential (LJ-13) (Jones, 1924), we adopt a Transformer-based architecture. Specifically, we opted for a non-equivariant Transformer architecture inspired by the Diffusion Transformer (DiT) Peebles & Xie (2022) supplemented by data augmentation. This architecture treats each particle as a token. Particle coordinates $x_i$ are embedded into a higher-dimensional space, and time $t$ and shortcut interval $d$ are encoded using sinusoidal embeddings (Vaswani et al., 2017) and processed to form the conditioning vector. Empirically, we found this non-equivariant transformer architecture to outperform equivariant network architectures such as EGNN (Satorras et al., 2021) and MACE-layer (Batatia et al., 2022) in terms of training efficiency and model performance. Full architectural specifications, including layer counts, hidden dimensions, activation functions, and specific input/output mappings, are provided in E.

## 5 Experiments

We evaluate the performance against baselines on various benchmark distributions. Detailed experimental & hyper-parameter settings and additional results are provided in Appendices E and F; below is a summary of the experiment settings:

- **Target densities**: We use two synthetic targets, GMM-40 and MW-32 (Midgley et al., 2023), for multi-modal and increased dimensionality settings. We also consider N-body system simulations by sampling from Double Well 4 (DW-4) and Lennard-Jones 13 (LJ-13) densities.

- **Baselines**: The baseline methods include: **FAB** (Midgley et al., 2023), **iDEM** (Sadegh et al., 2024), **LFIS** (Tian et al., 2024), **LIBD** (Máté & Fleuret, 2023) and flow-based samplers trained with the **PINN** objective in (Albergo & Vanden-Eijnden, 2025).

- **Metrics**: We use Wasserstein-2 distance ($\mathcal{W}_2$) and Total Variation (TV) distance computed between generated and ground-truth/high-quality MCMC samples. Metrics are computed either in the data space ($\mathcal{X}$) to assess sample configurations, energy space ($\mathcal{E}$) to assess distribution matching w.r.t. the potential, or interatomic distance space ($\mathcal{D}$) for N-body systems' physical invariances.

### 5.1 Synthetic Benchmarks

We test NFS$^2$ on two synthetic targets (Midgley et al., 2023): GMM-40, a 2D Gaussian Mixture Model with 40 well-separated modes testing exploration, and MW-32, a 32D potential with $2^{16}$ modes testing high-dimensional sampling. For both, we use the MLP architecture described in Section 4.4.

Table 1: Comparison of neural samplers on N-body systems (DW-4, LJ-13) and Synthetic Benchmarks (GMM-40, MW-32), with mean and standard deviation obtained from five independent runs.

| Energy → | DW-4 ($d = 8$) | | | LJ-13 ($d = 39$) | | | GMM-40 ($d = 2$) | | MW-32 ($d = 32$) | |
|---|---|---|---|---|---|---|---|---|---|---|
| Method ↓ | $\mathcal{E}\text{-}\mathcal{W}_2 \downarrow$ | $\mathcal{E}\text{-TV} \downarrow$ | $\mathcal{D}\text{-TV} \downarrow$ | $\mathcal{E}\text{-}\mathcal{W}_2$ | $\mathcal{E}\text{-TV}$ | $\mathcal{D}\text{-TV}$ | $\mathcal{E}\text{-}\mathcal{W}_2$ | $\mathcal{X}\text{-TV}$ | $\mathcal{E}\text{-TV}$ | $\mathcal{X}\text{-}\mathcal{W}_2$ |
| FAB | $0.64_{\pm 0.20}$ | $0.13_{\pm 0.01}$ | $0.07_{\pm 0.01}$ | $31.28_{\pm 0.31}$ | $0.94_{\pm 0.03}$ | $0.26_{\pm 0.01}$ | $8.89_{\pm 2.20}$ | $0.84_{\pm 0.19}$ | $0.17_{\pm 0.01}$ | $5.78_{\pm 0.02}$ |
| iDEM | $0.24_{\pm 0.11}$ | $0.12_{\pm 0.01}$ | $0.08_{\pm 0.01}$ | $13.13_{\pm 5.30}$ | $0.31_{\pm 0.01}$ | $0.03_{\pm 0.01}$ | $1.27_{\pm 0.21}$ | $0.83_{\pm 0.01}$ | $0.66_{\pm 0.15}$ | $8.18_{\pm 0.04}$ |
| LFIS | $5.22_{\pm 1.05}$ | $0.68_{\pm 0.02}$ | $0.29_{\pm 0.01}$ | $\infty$ | $*$ | $0.88_{\pm 0.00}$ | $0.27_{\pm 0.21}$ | $0.84_{\pm 0.01}$ | $*$ | $8.89_{\pm 0.03}$ |
| LIBD | $0.35_{\pm 0.01}$ | $0.14_{\pm 0.01}$ | $0.06_{\pm 0.01}$ | $49.89_{\pm 0.12}$ | $*$ | $0.45_{\pm 0.01}$ | $19.60_{\pm 0.26}$ | $0.83_{\pm 0.01}$ | $*$ | $8.62_{\pm 0.01}$ |
| PINN | $17.47_{\pm 21.8}$ | $0.20_{\pm 0.01}$ | $0.18_{\pm 0.01}$ | $48.3_{\pm 0.1}$ | $*$ | $0.44_{\pm 0.00}$ | $1.15_{\pm 0.12}$ | $0.73_{\pm 0.01}$ | $*$ | $8.32_{\pm 0.01}$ |
| NFS$^2$-128 (ours) | $1.03_{\pm 0.03}$ | $0.16_{\pm 0.01}$ | $0.07_{\pm 0.01}$ | $1.93_{\pm 0.01}$ | $0.19_{\pm 0.10}$ | $0.05_{\pm 0.00}$ | $0.12_{\pm 0.01}$ | $0.64_{\pm 0.00}$ | $0.30_{\pm 0.00}$ | $6.37_{\pm 0.01}$ |
| NFS$^2$-64 (ours) | $1.88_{\pm 0.16}$ | $0.19_{\pm 0.01}$ | $0.08_{\pm 0.01}$ | $1.51_{\pm 0.10}$ | $0.19_{\pm 0.10}$ | $0.06_{\pm 0.01}$ | $0.16_{\pm 0.01}$ | $0.65_{\pm 0.01}$ | $0.38_{\pm 0.01}$ | $6.47_{\pm 0.01}$ |
| NFS$^2$-32 (ours) | $3.91_{\pm 0.56}$ | $0.22_{\pm 0.01}$ | $0.11_{\pm 0.01}$ | $1.62_{\pm 0.20}$ | $0.20_{\pm 0.01}$ | $0.06_{\pm 0.00}$ | $0.17_{\pm 0.02}$ | $0.69_{\pm 0.01}$ | $0.54_{\pm 0.01}$ | $6.85_{\pm 0.01}$ |
| NFS$^2$-8 (ours) | $10.74_{\pm 0.51}$ | $0.44_{\pm 0.01}$ | $0.44_{\pm 0.01}$ | $29.57_{\pm 16.06}$ | $0.30_{\pm 0.01}$ | $0.22_{\pm 0.01}$ | $9.87_{\pm 0.01}$ | $0.73_{\pm 0.01}$ | $*$ | $13.91_{\pm 0.17}$ |

Note: $*$ in TV metrics indicates that the empirical sample distribution ($P_S$) and the ground truth distribution ($P_G$) have disjoint supports, i.e., $supp(P_S) \cap supp(P_G) = \emptyset$. This results in a meaningless Total Variation distance.

**Results.** Quantitative results are presented in Table Table 1. On GMM-40, NFS$^2$-128 achieves strong performance, particularly in data-space TV ($\mathcal{X}$-TV), indicating better mode coverage when compared to baselines (Figure 3). NFS$^2$-128 is on par with FAB and iDEM in energy-space $\mathcal{W}_2$ ($\mathcal{E}$-$\mathcal{W}_2$). While LFIS achieves the lowest $\mathcal{E}$-$\mathcal{W}_2$, its high $\mathcal{X}$-TV suggests potential issues with mode coverage.

On the high-dimensional MW-32 task, NFS$^2$-128 achieves a strong energy-space TV ($\mathcal{E}$-TV), ranking second after FAB, suggesting accurate energy distribution matching. It also performs competitively in data-space $\mathcal{W}_2$ ($\mathcal{X}$-$\mathcal{W}_2$). Visualisations in Figure 4 confirm that NFS$^2$, similar to FAB, accurately models the positions and the mixture weights of the modes, whereas LFIS, LIBD, and PINN struggle significantly and iDEM misrepresents mode mixture weights despite finding their locations. Moreover, the sample quality of the NFS$^2$ is well maintained when decreasing the number of integration steps during sampling, as further evidenced by our ablation study in Appendix F.

## 5.2 N-body System Simulations

**Double Well 4 (DW-4).** This system involves 4 particles in 2D governed by pairwise interactions ($d = 8$). We use the MLP architecture augmented with pairwise distance features (Section 4.4) and employ the data augmentation described in Appendix D. Due to the system's symmetries, we evaluate metrics invariant to it: energy-based metrics ($\mathcal{E}$-$\mathcal{W}_2$, $\mathcal{E}$-TV) and the TV distance between the distributions of sorted interatomic distances ($\mathcal{D}$-TV).

As shown in Table Table 1, NFS$^2$-128 remains competitive: it achieves a $\mathcal{D}$-TV of 0.07 which matches FAB, and an $\mathcal{E}$-TV of 0.16. It significantly outperforms LFIS across all metrics, indicating the effectiveness of amortization and improved $\partial_t \log Z_t$ estimates. In terms of $\mathcal{E}$-$\mathcal{W}_2$, iDEM and LIBD perform the best but NFS$^2$-128 is not too far from it. Importantly, halving the sampling budget from 128 to 64 steps results in only a marginal degradation in TV metrics and a slight increase in $\mathcal{E}$-$\mathcal{W}_2$. Figure 5 histograms show that NFS$^2$ captures both the bimodal interatomic distance and the energy distributions, comparable FAB, iDEM, and PINN, and notably better than LFIS.

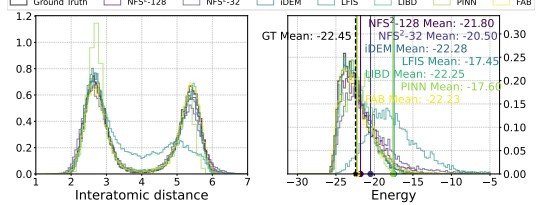

Figure 5: Histogram of interatomic distance and sample energy on DW-4.

**Lennard-Jones 13 (LJ-13).** This system involves 13 particles in 3D governed by pairwise interactions ($d = 39$). The LJ potential energy exhibits a high degree of non-linearity, making the sampling task significantly more challenging. We used the Transformer-based network and training method described in Section 4.4.

On the more complex LJ-13 system, NFS$^2$ demonstrates good performance, the first in PINN-based model to achieve this, as detailed in Table Table 1. Notably, NFS$^2$-64 achieves the best $\mathcal{E}$-$\mathcal{W}_2$ (1.51) among all methods, significantly outperforming iDEM and PINN; LFIS failed to produce a finite value for this metric (due to the asymptotic behaviour of the energy

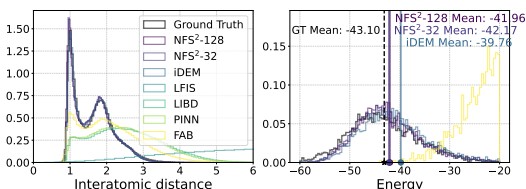

Figure 6: Histogram of interatomic distance and sample energy on LJ-13.

Figure 8: Visual comparison of shortcut regularisation methods on GMM-40.

Table 2: Comparison of Midpoint and Generalised shortcut consistency on GMM-40.

| Method | Consistency | $\mathcal{E}$-$\mathcal{W}_2 \downarrow$ | $\mathcal{X}$-TV $\downarrow$ |
|---|---|---|---|
| NFS$^2$-128 | Midpoint | $0.46_{\pm0.14}$ | $0.67_{\pm0.00}$ |
| NFS$^2$-128 | Generalised | $0.12_{\pm0.01}$ | $0.64_{\pm0.00}$ |
| NFS$^2$-64 | Midpoint | $1.32_{\pm0.29}$ | $0.69_{\pm0.01}$ |
| NFS$^2$-64 | Generalised | $0.16_{\pm0.01}$ | $0.65_{\pm0.01}$ |
| NFS$^2$-32 | Midpoint | $4.38_{\pm1.14}$ | $0.72_{\pm0.01}$ |
| NFS$^2$-32 | Generalised | $0.17_{\pm0.02}$ | $0.69_{\pm0.01}$ |
| NFS$^2$-8 | Midpoint | $20.29_{\pm2.72}$ | $0.84_{\pm0.01}$ |
| NFS$^2$-8 | Generalised | $9.87_{\pm0.01}$ | $0.73_{\pm0.01}$ |

function). All NFS$^2$ variants (from 128 to 32 steps) achieve superior $\mathcal{E}$-TV values compared to iDEM, while LFIS and PINN failed to converge for $\mathcal{E}$-TV. Specifically, NFS$^2$-64 maintains identical performance to NFS$^2$-128, whilst NFS$^2$-8 remains competitive with only 8 integration steps. The performance of NFS$^2$ in $\mathcal{D}$-TV degrades as expected with reduced sampling budget, from 0.05 for NFS$^2$-128 to 0.16 for NFS$^2$-32.

### 5.3 Ablation Studies

**Selection of $q(x)$ and model architecture.** We studied the impact of model architecture and data augmentation on performance for LJ-13. Models in comparison include: MLP, symmetry-equivariant EGNN (Satorras et al., 2021), DiT Transformer (Peebles & Xie, 2022), and DiT enhanced with symmetry-aware data augmentation plus additional residual-based resampling. As shown in Figure 7, the combination of the Transformer architecture and appropriate data augmentation yielded superior results. This highlights the benefit of leveraging both flexible architectures and symmetry information through augmentation, consistent with findings in related work (Peebles & Xie, 2022), highlighting the importance of architecture design for neural samplers.

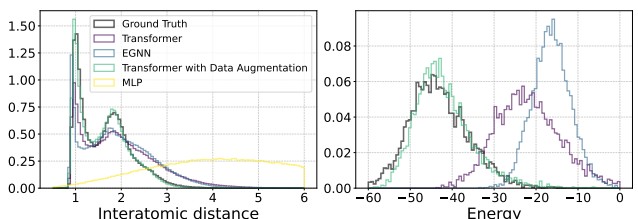

Figure 7: Histogram of interatomic distance and sample energy on LJ-13.

**The importance of good $\partial_t \log Z_t$ estimations.** The PINN baseline, which directly learns $\partial_t \log Z_t$ via gradient descent, performs well in low dimensional benchmarks, but fails in high dimensions (Table 1). This aligns with our experiences in developing NFS$^2$: inaccurate, high-variance $\partial_t \log Z_t$ estimations often caused diverged training. Figure 9 shows the average Mean Squared Error (MSE) of $\partial_t \log Z_t$ estimations versus ground truth in LJ-13 tests. The large error highlights the struggle of PINN in accurately learning $\partial_t \log Z_t$ in high dimensions. In short, accurate $\partial_t \log Z_t$ estimation is pivotal for many neural samplers' stability and training success.

**Shortcut regularisation enables faster sampling.** To reduce the sampling cost, NFS$^2$ is trained using additional trajectory shortcut consistency loss (Eq. (11)). We compared a model trained with the regularisation enabled against one trained without it. As shown qualitatively in Figure 8 for GMM-40, the regularisation effect is evident: the model trained without this consistency loss performs substantially worse as the number of sampling steps reduces.

**Generalised shortcut consistency loss further improves performance.** We compared the original shortcut consistency loss in Frans et al. (2024) which only considers the interval midpoint ("Midpoint")

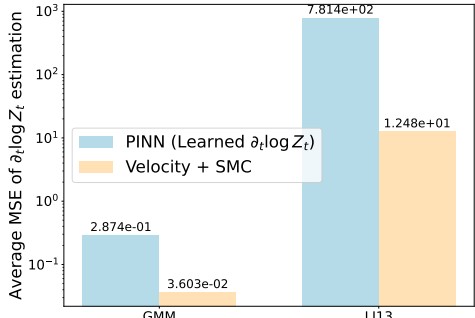

Figure 9: Comparison of $\partial_t \log Z_t$ estimates.

with our proposed loss in Eq. (11) with randomised interval segmentation ("Generalised"). Table 2 presents the results for GMM-40, where the generalised consistency approach yields better results across all tested step counts, particularly notable in the low budget domain. This means empirically, the generalised objective provides stronger regularisation regarding shortcut consistency.

**Better shortcut consistency leads to fewer steps required for high quality samples.** Empirically in GMM-40 experiments, stronger shortcut consistency regularisation substantially reduces the sampling budget to achieve comparable sample quality (measured by $\mathcal{X}$-TV in Figure 10). For instance, a strongly regularised $\text{NFS}^2$ sampler ($\lambda = 10$) with a 4-step sampling budget matches $\text{NFS}^2$ with no shortcut and 64 steps. Overall, while strong shortcut regularisation slightly hinders sample quality for $\text{NFS}^2$ samplers at high sampling budgets, it boosts performances significantly when considering low sampling budgets.

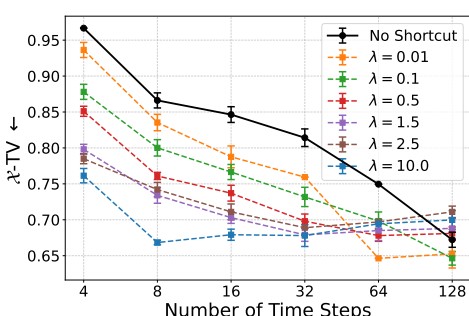

Figure 10: Comparison of $\mathcal{X}$-TV across different shortcut regularisation strength $\lambda$.

## 6    CONCLUSIONS AND LIMITATIONS

In this paper, we proposed Neural Flow Shortcut Samplers ($\text{NFS}^2$), a flow-based sampler that is straightforward to train and offers dynamic adjustability in its sampling budget. $\text{NFS}^2$ achieves competitive performance to contemporary state-of-the-art approaches across key benchmark tasks. This is particularly evident in its ability to maintain high sample quality even with significantly reduced simulation steps.

Despite these advancements, $\text{NFS}^2$ has limitations. Firstly, a limitation of the entire class of neural samplers (not just $\text{NFS}^2$) is the need for nontrivial training compared to MCMC; advantages are clearest in amortized sampling settings where a trained sampler transfers to related targets without retraining. Secondly, the required computations can be heavy: the divergence term scales poorly in high dimensions, and estimating $\partial_t \log Z_t$ remains intractable—especially for large particle systems (e.g., Lennard–Jones–155). Further investigation is warranted to understand the full implications of these challenges and to explore potential mitigation strategies. Such strategies could include the use of stochastic divergence estimation (Hutchinson, 1989) and employing auxiliary variable methods. We discuss additional limitations and potential avenues for future work in more detail in Appendix G.

## 7    ETHICS STATEMENT

This paper presents work whose goal is to advance machine learning research. There may exist potential societal consequences of our work; however, none of which we feel must be specifically highlighted here at the moment of paper submission.

## 8    REPRODUCIBILITY STATEMENT

We would release all applicable source code and trained model weights to enable full reproducibility of our results. A self-contained Jupyter Notebook is provided in the supplementary materials; it includes end-to-end scripts for preparation, training, and sampling.

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

# Appendix for "Neural Flow Samplers with Shortcut Models"

## CONTENTS

## A   IMPORTANCE SAMPLING AND SEQUENTIAL MONTE CARLO

This section reviews the basic Sequential Monte Carlo (SMC) algorithm. We begin by introducing importance sampling and its application to estimating the intractable time derivative $\partial_t \log Z_t$, as presented in Tian et al. (2024). We then proceed with an introduction to Sequential Monte Carlo, which is employed in our methods to estimate $\partial_t \log Z_t$.

### A.1   IMPORTANCE SAMPLING

Consider a target distribution $\pi(x) = \frac{\rho(x)}{Z}$, where $\rho(x) \geq 0$ is the unnormalised probability density and $Z = \int \rho(x) \, \mathrm{d}x$ denotes the normalising constant, which is typically intractable. For a test function $\phi(x)$ of interest, estimating its expectation under $\pi$ through direct sampling can be challenging. Importance sampling (IS) (Kahn, 1950) instead introduces a proposal distribution $q$, which is easy to

sample from, and proposes an expectation estimator as follows

$$\mathbb{E}_{\pi(x)}[\phi(x)] = \frac{1}{Z}\mathbb{E}_{q(x)}\left[\frac{\rho(x)}{q(x)}\phi(x)\right] = \frac{\mathbb{E}_{q(x)}\left[\frac{\rho(x)}{q(x)}\phi(x)\right]}{\mathbb{E}_{q(x)}\left[\frac{\rho(x)}{q(x)}\right]}. \tag{12}$$

Thus, the expectation can be estimated via the Monte Carlo method

$$\mathbb{E}_{\pi(x)}[\phi(x)] \approx \sum_{k=1}^{K} \frac{w^{(k)}}{\sum_{=1}^{N} w^{(j)}}\phi(x^{(k)}), \quad x^{(k)} \sim q(x), \tag{13}$$

where $w^{(k)} = \frac{\rho(x^{(k)})}{q(x^{(k)})}$ denotes the importance weight. While importance sampling yields a consistent estimator as $N \to \infty$, it typically suffers from high variance and low effective sample size (Thiébaux & Zwiers, 1984) when the proposal deviates from the target distribution. In theory, a zero-variance estimator can be achieved if $q(x) \propto \rho(x)\phi(x)$; however, this condition is rarely satisfied in practice. This limitation renders importance sampling inefficient in high-dimensional spaces, as a large number of Monte Carlo samples are required to mitigate the variance.

**Approximating $\partial_t \log Z_t$ with Importance Sampling.** Tian et al. (2024) propose approximating $\partial_t \log Z_t$ using importance sampling, where they express $\partial_t \log Z_t \approx \sum_k \frac{w_t^{(k)}}{\sum_k w_t^{(k)}}\partial_t \log \tilde{p}_t(x_t^{(k)})$. Here $x_t^{(k)} \sim p_t(x;\theta)$ denotes the sample generated by the velocity $v_t(x;\theta)$ at time $t$, and $\log w_t^{(k)} = \int_0^t \delta_\tau(x_\tau; v_t(\cdot;\theta)) \, d\tau$. For completeness, we provide a step-by-step recall of the proof of the correctness of this estimator from Tian et al. (2024).

First, we show that $\partial_t \log Z_t$ is given by the expectation over $p_t$:

$$\partial_t \log Z_t = \partial_t \log \int \tilde{p}_t(x) \, dx = \frac{1}{Z_t}\int \tilde{p}_t(x)\partial_t \log \tilde{p}_t(x) \, dx = \mathbb{E}_{p_t(x)}[\partial_t \log \tilde{p}_t(x)]. \tag{14}$$

$\partial_t \log Z_t$ therefore can be estimated via importance sampling

$$\partial_t \log Z_t \approx \sum_{k=1}^{K} \frac{w_t^{(k)}}{\sum_j w_t^{(j)}}\partial_t \log \tilde{p}_t(x^{(k)}), \quad x^{(k)} \sim p_t(x;\theta), \tag{15}$$

where $w_t^{(k)} = \frac{p_t(x^{(k)})}{p_t(x^{(k)};\theta)}$. Next, we show that the weight $\log w_t^{(k)}$ is given by integrating $\delta_t(x; v_t(\cdot;\theta))$ on $[0, t]$, where $\delta_t$ is defined in Equation (6) i.e. $\log \frac{p_t(x_t)}{p_t(x_t;\theta)} = \int_0^t \delta_s(x; v_s(\cdot;\theta)) \, ds$.

We begin by computing the instantaneous rate of change of the log-densities of the model $p_t(x_t;\theta)$ along the trajectories generated by $v_t(x_t;\theta)$, as in the instantaneous change of variable formula in Equation (4)

$$\begin{aligned}\partial_t[\log p_t(x_t;\theta)] &= \partial_t \log p_t(x_t;\theta) + \nabla_{x_t} \log p_t(x_t;\theta) \cdot \frac{dx_t}{dt} \\ &= (-\nabla_{x_t} \cdot v_t(x_t;\theta) - v_t(x_t;\theta) \cdot \nabla_{x_t} \log p_t(x_t;\theta)) + \nabla_{x_t} \log p_t(x_t;\theta) \cdot v_t(x_t;\theta) \\ &= -\nabla_{x_t} \cdot v_t(x_t;\theta). \end{aligned} \tag{16}$$

Similarly, the instantaneous rate of change of the log-densities of the target $p_t(x_t)$ along the trajectories generated by $v_t(x_t;\theta)$ is

$$\begin{aligned}\partial_t[\log p_t(x_t)] &= \partial_t \log p_t(x_t) + \nabla_{x_t} \log p_t(x_t) \cdot \frac{dx_t}{dt} \\ &= \partial_t \log p_t(x_t) + \nabla_{x_t} \log p_t(x_t) \cdot v_t(x_t;\theta). \end{aligned} \tag{17}$$

Note that the score of the target is tractable $\nabla_x \log p_t(x) = \nabla_x \log \tilde{p}_t(x)$. Therefore, the log densities along the trajectories can be computed via

$$\log p_t(x_t;\theta) = \log p_0(x_0;\theta) - \int_0^t \nabla_{x_s} \cdot v_s(x_s;\theta) \, ds \tag{18}$$

$$\log p_t(x_t) = \log p_0(x_0) + \int_0^t [\partial_s \log p_s(x_s) + \nabla_{x_s} \log p_s(x_s) \cdot v_s(x_s;\theta)] \, ds. \tag{19}$$

Since $p_0(x; \theta) = p_0(x) = \eta(x)$ due to the annealing path construction $p_t \propto \rho^t \eta^{1-t}$, we have

$$\log \frac{p_t(x_t)}{p_t(x_t; \theta)} = \int_0^t [\nabla_{x_s} \cdot v_s(x_s; \theta) + \partial_s \log p_s(x_s) + \nabla_{x_s} \log p_s(x_s) \cdot v_s(x_s; \theta)] \, \mathrm{d}s$$

$$= \int_0^t \delta_s(x; v_s(\cdot; \theta)) \, \mathrm{d}s, \tag{20}$$

which completes the proof.

**Remark.** Although importance sampling offers an elegant method for estimating the intractable time derivatives $\partial_t \log Z_t$, it faces two main challenges. First, as previously discussed, importance sampling typically suffers from high variance and requires large sample sizes to improve the effective sample size. More critically, the computation of the weight involves the intractable term $\partial_t \log p_s(x_t)$, which in turn depends on $\partial_t \log Z_t = \mathbb{E}_{p_t(x)}[\partial_t \log \tilde{p}_t(x)]$. Tian et al. (2024) approximate this expectation naively by averaging $\partial_t \log \tilde{p}_t(x)$ over the mini-batch during training, which introduces both approximation errors and bias in the importance weights.

In the following section, we introduce Sequential Monte Carlo, which is employed in our methodology to estimate $\partial_t \log Z_t$, balancing the efficiency of the short-run MCMC driven by the velocity with the effectiveness of low variance.

## A.2 SEQUENTIAL MONTE CARLO

Sequential Monte Carlo (SMC) provides an alternative method to estimate the intractable expectation $\mathbb{E}_{p_t(x)}[\phi(x)]$. Specifically, SMC decomposes the task into easier subproblems involving a set of unnormalised intermediate target distributions $\{\tilde{p}_{t_m}(x_{t_m})\}_{m=0}^M$.[1] We begin by introducing sequential importance sampling (SIS):

$$\mathbb{E}_{p_{t_m}(x)}[\phi(x)] = \int q(x_{t_0:t_m}) \frac{p(x_{t_0:t_m})}{q(x_{t_0:t_m})} \phi(x_{t_m}) \, \mathrm{d}x_{t_0:t_m}$$

$$\approx \frac{1}{K} \sum_{k=1}^K \frac{p(x_{t_0:t_m}^{(k)})}{q(x_{t_0:t_m}^{(k)})} \phi(x_{t_m}^{(k)}), \quad \text{where} \quad x_{t_0:t_m}^{(k)} \sim q(x_{t_0:t_m}^{(k)}). \tag{21}$$

The importance weights are $w_{t_m}^{(k)} = \frac{p(x_{t_0:t_m}^{(k)})}{q(x_{t_0:t_m}^{(k)})}$. The key ingredients of SMC are the proposal distributions $q(x_{t_0:t_m})$ and the target distributions $p(x_{t_0:t_m})$. Here we consider a general form associated with a sequence of forward kernels $q(x_{t_0:t_m}) = q(x_{t_0}) \prod_{s=0}^{m-1} \mathcal{F}_{t_s}(x_{t_{s+1}}|x_{t_s})$, and the target distribution is defined by a sequence of backward kernels $p(x_{t_0:t_m}) = p(x_{t_m}) \prod_{s=0}^{m-1} \mathcal{B}_{t_s}(x_{t_s}|x_{t_{s+1}})$. Substituting this into the expression for the importance weights gives

$$w_{t_m}^{(k)} = \frac{p(x_{t_0:t_m}^{(k)})}{q(x_{t_0:t_m}^{(k)})} = \frac{p(x_{t_m}) \prod_{s=0}^{m-1} \mathcal{B}_{t_s}(x_{t_s}|x_{t_{s+1}})}{q(x_{t_0}) \prod_{s=0}^{m-1} \mathcal{F}_{t_s}(x_{t_{s+1}}|x_{t_s})} = w_{t_{m-1}}^{(k)} W_{t_m}^{(k)}, \tag{22}$$

where $W_{t_m}^{(k)}$, termed the incremental weights, are calculated as,

$$W_{t_m}^{(k)} = \frac{p_{t_m}(x_{t_m})}{p_{t_{m-1}}(x_{t_{m-1}})} \frac{\mathcal{B}_{t_m}(x_{t_{m-1}}|x_{t_m})}{\mathcal{F}_{t_m}(x_{t_m}|x_{t_{m-1}})}. \tag{23}$$

By defining the backward kernel as $\mathcal{B}_{t_m}(x_{t_{m-1}}|x_{t_m}) = \frac{p_{t_{m-1}}(x_{t_{m-1}}) \mathcal{F}_{t_m}(x_{t_m}|x_{t_{m-1}})}{p_{t_{m-1}}(x_{t_m})}$, the incremental weight is tractable and becomes

$$W_{t_m}^{(k)} = \frac{p_{t_m}(x_{t_m})}{p_{t_{m-1}}(x_{t_m})}. \tag{24}$$

Therefore, the expectation can be approximated as

$$\mathbb{E}_{p_{t_m}(x)}[\phi(x)] \approx \sum_k \tilde{w}_{t_m}^{(k)} \phi(x_{t_m}), \quad \tilde{w}_{t_m}^{(k)} = \frac{w_{t_m}^{(k)}}{\sum_j w_{t_m}^{(j)}}, \quad w_{t_m}^{(k)} = w_{t_{m-1}}^{(k)} W_{t_m}^{(k)} \propto w_{t_{m-1}}^{(k)} \frac{\tilde{p}_{t_m}(x_t)}{\tilde{p}_{t_{m-1}}(x_{t_m})}.$$

---

[1] We consider a discrete-time schedule that satisfies $0 = t_0 < t_1 < \cdots < t_m < \cdots < t_{M-1} < t_M = 1$.

The SIS method is elegant, as the weights can be computed on the fly. However, with a straightforward application of SIS, the distribution of importance weights typically becomes increasingly skewed as $t$ progresses, resulting in many samples having negligible weights. This imbalance reduces the effective sample size and the overall efficiency of the algorithm. To alleviate this issue, a common approach used in SMC is to introduce a resampling step. At each time step $t$, the samples $x_t^{(k)}$ are resampled using systematic resampling method based on the normalized weights $\tilde{w}_t^{(k)}$[2].

The resampled particles are then assigned equal weights to mitigate the bias introduced by the skewness in the weight distribution. This resampling trick prevents the sample set from degenerating, where only a few particles carry significant weight while others contribute minimally. By periodically resampling, the algorithm maintains diversity in the particle set. It ensures that the estimation is focused on regions of high probability density, leading to less skewed importance weight distributions. To encourage the convergence of MCMC transition kernels, we also introduce a velocity-driven step. The implementation of the proposed velocity-driven sequential Monte Carlo (VD-SMC) is given by Algorithm 2. Given the sample size $K$ and the time schedule $\{t_m\}_{m=1}^M$ with $t_0 = 0, t_M = 1$, the algorithm VD-SMC returns the samples and importance weights $\{x_{t_m}^{(k)}, \tilde{w}_{t_m}^{(k)}\}_{k=1,m=0}^{K,M}$. Therefore, the intractable time derivative can be approximated as $\partial_t \log Z_t = \mathbb{E}_{p_t} \partial_t \log \tilde{p}_t(x) \approx \sum_k \tilde{w}_t^{(k)} \partial_t \log \tilde{p}_t(x_t^{(k)})$. However, as illustrated in Figure 2, the estimation of $\mathbb{E}_{p_t} \partial_t \log \tilde{p}_t(x)$ exhibits higher variance compared to using $\mathbb{E}_{p_t} \xi_t(x; v_t)$. There-

---

**Algorithm 2** Velocity-Driven SMC

**Input**: velocity $v_t(\cdot; \theta)$; sample size $K$; time steps $\{t_m\}_{m=0}^M$
**Output**: samples and weights $\{x_{t_m}^{(k)}, \tilde{w}_{t_m}^{(k)}\}_{k=1,m=0}^{K,M}$

1: **procedure** VD-SMC($v_t, K, \{t_m\}_{m=0}^M$)
2:     **for** $k = 1, \ldots, K$ **do**
3:         $x_0^{(k)} \sim p_0(x_0), \quad w_0^{(k)} = p_0(x_0^{(k)})$
4:     **end for**
5:     $\tilde{w}_0^{(k)} = w_0^{(k)} / \sum_{i=1}^K w_0^{(i)}$
6:     **for** $m = 1, \ldots, M$ **do**
7:         **for** $k = 1, \ldots, K$ **do**
8:             **if** $\text{ESS}(\tilde{w}_{t_{m-1}}^{1:K}) < \text{ESS}_{\min}$ **then**
9:                 $a_{t_m}^{(k)} \sim \text{Systematic}(\tilde{w}_{t_{m-1}}^{1:K}), \hat{w}_{t_{m-1}}^{(k)} = 1$
10:           **else**
11:              $a_{t_m}^{(k)} = k, \quad \hat{w}_{t_{m-1}}^{(k)} = w_{t_{m-1}}^{(k)}$
12:           **end if**
13:           $\text{d}t \leftarrow t_m - t_{m-1}$
14:           $x_{t_m}^{(k)} = \text{HMC}(x_{t_{m-1}}^{(a_{t_m}^{(k)})} + \eta v_{t_{m-1}}(x_{t_{m-1}}^{(a_{t_m}^{(k)})}; \theta) \, \text{d}t)$
15:           $w_{t_m}^{(k)} = \hat{w}_{t_{m-1}}^{(k)} \frac{\tilde{p}_{t_m}(x_{t_m}^{(k)})}{\tilde{p}_{t_{m-1}}(x_{t_m}^{(k)})}$
16:         **end for**
17:         $\tilde{w}_{t_m} = w_{t_m}^{(k)} / \sum_{i=1}^K w_{t_m}^{(i)}$
18:     **end for**
19: **end procedure**

---

fore, in practice, we approximate the time derivative as $\partial_t \log Z_t = \mathbb{E}_{p_t} \xi_t(x; v_t(\cdot; \theta_{\text{sg}})) \approx \sum_k \tilde{w}_t^{(k)} \xi_t(x_t^{(k)}; v_t(\cdot; \theta_{\text{sg}}))$, where $\theta_{\text{sg}}$ denotes the parameters with gradients detached.

## B    VARIANCE REDUCTION WITH CONTROL VARIATES

### B.1    PROOF OF EQUATION (9)

Recall that in Equation (9), we show that the following equation holds:

$$\partial_t \log Z_t = \underset{c_t}{\text{argmin}} \, \mathbb{E}_{p_t} (\xi_t(x; v_t) - c_t)^2, \quad \xi_t(x; v_t) \triangleq \partial_t \log \tilde{p}_t(x) + \nabla_x \cdot v_t(x) + v_t(x) \cdot \nabla_x \log p_t(x).$$

We provide a detailed proof of this result here. First, we have the following Lemmas.

**Lemma 1** (Stein's Identity (Stein et al., 2004)). *Assuming that the target density $p_t$ vanishes at infinity, i.e., $p_t(x) = 0$, whenever $\exists d$ such that $x[d] = \infty$, where $x[d]$ denotes the $d$-th element of the vector $x$. Under this assumption, we have the result: $\int [\nabla_x \cdot v_t(x) + v_t(x) \cdot \nabla_x \log p_t(x)] \tilde{p}_t(x) \, \text{d}x = 0$.*

---

[2]Rather than resampling at every time step $t$, a more advanced resampling method involves making the resampling decision adaptively, based on criteria such as the Effective Sample Size (Doucet et al., 2001).

*Proof.* To prove the result, notice that

$$\int [\nabla_x \cdot v_t(x) + v_t(x) \cdot \nabla_x \log p_t(x)] \tilde{p}_t(x) \, \mathrm{d}x = \int \tilde{p}_t(x) \nabla_x \cdot v_t(x) + v_t(x) \cdot \nabla_x \tilde{p}_t(x) \, \mathrm{d}x$$

$$= \int \nabla_x \cdot [v_t(x) \tilde{p}_t(x)] \, \mathrm{d}x$$

$$= \sum_d \int \frac{\mathrm{d}}{\mathrm{d}x_d} [v_t(x) \tilde{p}_t(x)][d] \, \mathrm{d}x_d$$

$$= \sum_d [v_t(x) \tilde{p}_t(x)][d]|_{x_d=-infty}^{x_d=+\infty} = 0,$$

where the last row applies the divergence theorem $\int_a^b f'(t) \, \mathrm{d}t = f(b) - f(a)$. □

**Lemma 2.** *Let* $c_t^* = \mathrm{argmin}_{c_t} \mathbb{E}_{p_t}(\xi_t(x; v_t) - c_t)^2$, *then* $c_t^* = \mathbb{E}_{p_t} \xi_t(x; v_t)$.

*Proof.* To see this, we can expand the objective

$$\mathcal{L}(c_t) = \mathbb{E}_{p_t}(\xi_t(x; v_t) - c_t)^2 = c_t^2 - 2c_t \mathbb{E}_{p_t} \xi_t(x; v_t) + c = (c_t - \mathbb{E}_{p_t} \xi_t(x; v_t))^2 + c',$$

where $c, c'$ are constants w.r.t. $c_t$. Therefore $c_t^* = \mathrm{argmin}_{c_t} \mathbb{E}_{p_t}(\xi_t(x; v_t) - c_t)^2 = \mathbb{E}_{p_t} \xi_t(x; v_t)$. □

Now, it is ready to prove Equation (9). Specifically,

$$c_t^* = \mathbb{E}_{p_t} \xi_t(x; v_t)$$

$$= \frac{1}{\int \tilde{p}_t(x) \, \mathrm{d}x} \int \tilde{p}_t(x) [\partial_t \log \tilde{p}_t(x) + \nabla_x \cdot v_t(x) + v_t(x) \cdot \nabla_x \log p_t(x)] \, \mathrm{d}x$$

$$= \frac{1}{\int \tilde{p}_t(x) \, \mathrm{d}x} \int \partial_t \tilde{p}_t(x) + [\nabla_x \cdot v_t(x) + v_t(x) \cdot \nabla_x \log p_t(x)] \tilde{p}_t(x) \, \mathrm{d}x$$

$$= \frac{1}{\int \tilde{p}_t(x) \, \mathrm{d}x} \int \partial_t \tilde{p}_t(x) \, \mathrm{d}x$$

$$= \partial_t \log Z_t,$$

where the first and fourth equations follow Lemmas 1 and 2, respectively, which completes the proof.

**Remark.** Equation (9) provides an alternative approach to estimate $\partial_t \log Z_t$. As illustrated in Figure 2, this estimation exhibits lower variance compared to using $\mathbb{E}_{p_t} \partial_t \log \tilde{p}_t(x)$. This reduction in variance can potentially lead to better optimisation. To evaluate this, we conducted experiments on GMM datasets by minimizing the loss in Equation (6), employing two different methods to estimate $\partial_t \log Z_t$: $\mathbb{E}_{p_t} \partial_t \log \tilde{p}_t(x)$ and $\mathbb{E}_{p_t} \xi_t(x; v_t)$. The loss values during training are plotted against the training steps in Figure 11. The results show that the estimator of $\mathbb{E}_{p_t} \xi_t(x; v_t)$ achieves lower loss values, highlighting the superior training effects achieved with the lower variance estimation of $\partial_t \log Z_t$.

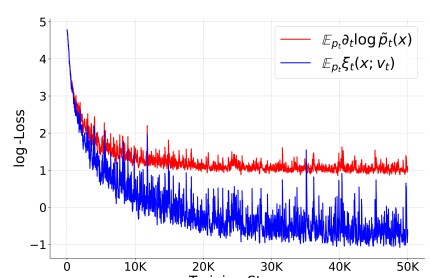

Figure 11: Training loss of using different estimators of $\partial_t \log Z_t$.

## B.2 STEIN CONTROL VARIATES

In this section, we provide a perspective from control variates to explain the observation of variance reduction in Figure 2. In particular, consider a Monte Carlo integration problem $\mu = \mathbb{E}_\pi[f(x)]$, which can be estimated as $\hat{\mu} = \frac{1}{K} \sum_{k=1}^K f(x^{(k)}), x^{(k)} \sim \pi$. Assuming another function exists with a known mean $\gamma = \mathbb{E}_\pi[g(x)]$, we call $g$ the control variate. We then can construct another estimator $\check{\mu} = \frac{1}{K} \sum_{k=1}^K (f(x^{(k)}) - \beta g(x^{(k)})) + \beta \gamma$, where $\beta$ is a scalar coefficient and controls the scale of

---

**Algorithm 3** Training Procedure of NFS$^2$ (one training epoch only for illustration)

---

**Input**: initial shortcut model $s_t(\cdot, \cdot; \theta)$, time spans $\{t_m\}_{m=0}^M$, regularisation weight $\lambda$, learning rate $\eta$
**Output**: trained shortcut model $s_t(\cdot, \cdot; \theta)$

1:  $\tilde{t}_0 \leftarrow 0, \tilde{t}_m \sim \mathcal{U}([t_{m-1}, t_m]), m = 1, \ldots, M$                         ▷ Sample time steps
2:  $\{x_{\tilde{t}_m}^{(k)}, \tilde{w}_{\tilde{t}_m}^{(k)}\}_{k=1,m=0}^{K,M} \leftarrow \text{VD} - \text{SMC}(s_t(\cdot, 0; \theta), K, \{\tilde{t}_m\}_{m=0}^M)$      ▷ Generate samples using
    Algorithm 2
3:  **for** $t, \{x_t, \tilde{w}_t\} \sim \{x_{\tilde{t}_m}^{(k)}, \tilde{w}_{\tilde{t}_m}^{(k)}\}_{k=1,m=0}^{K,M}$ **do**    ▷ Executed with mini-batch in parallel practically
4:     $\xi_t \leftarrow \partial_t \log \tilde{p}_t(x_t) + \nabla_x \cdot s_t(x_t, 0; \theta) + s_t(x_t, 0; \theta) \cdot \nabla_x \log p_t(x_t)$
5:     $c_t \leftarrow \sum_k \tilde{w}_t^{(k)} \left[ \partial_t \log \tilde{p}_t(x_t^{(k)}) + \nabla_x \cdot s_t(x_t^{(k)}, 0; \theta_{\text{sg}}) + s_t(x_t^{(k)}, 0; \theta_{\text{sg}}) \cdot \nabla_x \log p_t(x_t^{(k)}) \right]$   ▷ Estimate
    $\partial_t \log Z_t$ using Stein control variate
6:     $d \sim \mathcal{U}(0, 1)$                                             ▷ Sample shortcut distance
7:     $\alpha \sim \mathcal{U}(0, 1)$                               ▷ Sample random fraction for interval split
8:     $x_{t+\alpha d} \leftarrow x_t + s_t(x_t, \alpha d; \theta_{\text{sg}}) \alpha d$    ▷ Compute intermediate state using $s_t$ with stop-gradient
9:     $s_{\text{target}} \leftarrow \alpha s_t(x_t, \alpha d; \theta_{\text{sg}}) + (1 - \alpha) s_{t+\alpha d}(x_{t+\alpha d}, (1 - \alpha)d; \theta_{\text{sg}})$   ▷ Compute shortcut target
    with stop-gradient
10:    $\mathcal{L}(\theta) \leftarrow (\xi_t - c_t)^2 + \lambda \| s_t(x_t, d; \theta) - s_{\text{target}} \|_2^2$    ▷ Compute training loss (residual +
    consistency)
11:    $\theta \leftarrow \theta - \eta \nabla_\theta \mathcal{L}(\theta)$                            ▷ Perform gradient update
12: **end for**

---

the control variate. It is obvious that $\mathbb{E}[\check{\mu}] = \mathbb{E}[\hat{\mu}] = \mu, \forall \beta \in \mathbb{R}$. Moreover, we can choose a $\beta$ to minimize the variance of $\check{\mu}$. To obtain it, we first derive the variance of $\check{\mu}$

$$\mathbb{V}[\check{\mu}] = \frac{1}{K} (\mathbb{V}[f] - 2\beta \text{Cov}(f, g) + \beta^2 \mathbb{V}[g]). \tag{25}$$

Since $\mathbb{V}[\check{\mu}]$ is convex w.r.t. $\beta$, by differentiating it w.r.t. $\beta$ and zeroing it, we find the optimal value, $\beta^* = \text{Cov}(f, g)/\mathbb{V}[g]$. Substituting it into Equation (25), we get the minimal variance

$$\mathbb{V}[\check{\mu}] = \frac{1}{K} \mathbb{V}[\hat{\mu}] (1 - \text{Corr}(f, g)^2). \tag{26}$$

This shows that, given the optimal value $\beta^*$, any function $g$ that correlates to $f$, whether positively or negatively, reduces the variance of the estimator, i.e., $\mathbb{V}[\check{\mu}] < \mathbb{V}[\hat{\mu}]$. In practice, the optimal $\beta^*$ can be estimated from a small number of samples (Ranganath et al., 2014). However, the primary challenge lies in finding an appropriate function $g$. For a detailed discussion on control variates, see Geffner & Domke (2018).

Fortunately, Lemma 1 offers a systematic way to construct a control variate for $\mathbb{E}_{p_t}[f(x)] \triangleq \mathbb{E}_{p_t}[\partial_t \log \tilde{p}_t(x)] \approx \frac{1}{K} \sum_{k=1}^K \partial_t \log \tilde{p}_t(x^{(k)})$, where $x^{(k)} \sim p_t$. Specifically, we define $g(x) = \nabla_x \cdot v_t(x; \theta) + v_t(x; \theta) \cdot \nabla_x \log p_t(x)$, from which we have $\gamma = \mathbb{E}_{p_t}[g(x)] = 0$. Using this, we construct a new estimator:

$$\check{\mu} = \frac{1}{K} \sum_{k=1}^K \partial_t \log \tilde{p}_t(x^{(k)}) + \beta^* (\nabla_x \cdot v_t(x^{(k)}; \theta) + v_t(x^{(k)}; \theta) \cdot \nabla_x \log p_t(x^{(k)})), \quad x^{(k)} \sim p_t. \tag{27}$$

Moreover, when $\theta$ is optimal, Equation (6) equals zero, implying $g(x) = -f(x) + c$, where $c$ is a constant independent of the sample $x$. In this case, $\text{Corr}(f, g) = -1$, and $\check{\mu}$ becomes a zero-variance estimator. As an additional clarification, Stein's identity from Lemma 1 is also employed as a control variate in Liu et al. (2017), where it is utilized to optimise the policy in reinforcement learning.

## C   Training and Sampling Algorithms

The training and sampling algorithms are detailed in Algorithms 3 and 4, respectively. For clarity, Algorithm 3 illustrates a single training epoch. In particular, we parameterise the model with a single neural network $s_t(x, d; \theta)$ that takes the sample $x$, time step $t$, and shortcut distance $d$ as input to anticipate the shortcut direction. This design enables NFS$^2$ to model in continuous time, unlike

---

**Algorithm 4** Sampling Procedure of NFS[2]

---

**Input**: trained shortcut model $s_t(\cdot, \cdot; \theta)$, initial density $p_0$, # steps $M$
**Output**: generated samples $x$

1: $x_0 \sim p_0, \quad d \leftarrow \frac{1}{M}, \quad t \leftarrow 0$                                     ▷ Initialisation
2: **for** $m = 0, \ldots, M - 1$ **do**
3:      $x \leftarrow x + s_t(x, d; \theta)d$
4:      $t \leftarrow t + d$
5: **end for**

---

the baseline LFIS (Tian et al., 2024), which trains separate neural networks for each time step — a memory-intensive and inefficient approach. To train the model, we define time spans $\{t_m\}_{m=0}^{M}$ that are evenly distributed over $[0, 1]$, satisfying $0 = t_0 < \cdots < t_M = 1$ and $2t_m = t_{m+1} + t_{m-1}, \forall m$. In each epoch, we uniformly sample time steps from the time spans $\tilde{t}_m \sim \mathcal{U}([t_{m-1}, t_m])$ and ensure that $\tilde{t}_0 = 0$.[3] Subsequently, Algorithm 2 is invoked to generate training samples, as detailed in Appendix D. Notably, any $q$ distribution can be used to generate training samples. The choice of the proposed velocity-driven SMC is motivated by two key reasons:

i) At the beginning of training, the generated samples are far from the mode, encouraging the model to focus on mode-covering. As training progresses, the generated samples become more accurate, gradually shifting toward mode-seeking, ultimately balancing exploration and exploitation for improved learning efficiency.

ii) Improved $\partial_t \log Z_t$ estimation efficiency. SMC returns the samples and importance weights for each time step simultaneously, streamlining the estimation of $\partial_t \log Z_t$.

After generating training samples, we compute the loss in Equation (11) and update the model using gradient descent, as outlined in steps 4–9 of Algorithm 3.

## D    SELECTION OF $q_t(x)$ AND DATA AUGMENTATION

Selecting a $q_t(x)$ distribution is equivalent to generating a list of points at which the loss is calculated and from which backpropagation is conducted. Therefore, this selection is an important aspect of the training dynamics and has been an area of ongoing research in PINN-based learning procedures (Wu et al., 2023). Empirically, we conducted preliminary experiments with several approaches for generating training data points: (i) sampling uniformly in the $\mathbb{R}^d$ space; (ii) using an approximate $p_t(x)$ distribution obtained from SMC procedures; and (iii) combining data points from SMC procedures with random perturbations. For symmetry-aware systems, such as DW-4 and LJ-13, approach (iii) was further augmented with random rotations and translations along a randomly sampled axis.

We found that this augmented approach (iii) contributed to the observed good performance. Our conjecture is that uniform random sampling (i) is too inefficient, while using points solely from SMC (ii) lacks diversity because only modal regions are predominantly covered in the loss calculation, which may not be ideal.

Additionally, we employ a slight modification of the RAR-D resampling procedure (Wu et al., 2023). The basic idea is that after calculating the residual for a batch of points, we perform an additional learning step using points resampled from the same batch, where the empirical resampling distribution is determined by the residuals. That is, we focus our computational efforts more heavily on locations where our model exhibits higher residuals.

## E    EXPERIMENTAL DETAILS

### E.1    DATASETS

**Gaussian Mixture Model (GMM-40).** We use a 40 Gaussian mixture density in 2 dimensions as proposed by Midgley et al. (2023). This density consists of a mixture of 40 evenly weighted

---

[3]More advanced schedule beyond uniform sampling remain important future works.

Gaussians with identical covariances

$$\Sigma = \begin{bmatrix} 40 & 0 \\ 0 & 40 \end{bmatrix}$$

and $\mu_i$ are uniformly distributed over the $[-40, 40]$ box, i.e., $\mu_i \sim U(-40, 40)^2$.

$$p(x) = \frac{1}{40} \sum_{i=1}^{40} \mathcal{N}(x; \mu_i, \Sigma)$$

**Many Well 32 (MW-32).** We use a 32-dimensional Many Well density, as proposed by Midgley et al. (2023). This density consists of a mixture of $n_{\text{wells}} = 16$ independent double-well potentials:

$$E(x) = \sum_{i=1}^{n_{\text{wells}}} E_{\text{DW}}(x_i)$$

where each $x_i$ corresponds to a pair of variables in a 2-dimensional space. The unnormalized log density for a single 2D Double Well is:

$$\log p_{\text{DW}}(x_1, x_2) = -x_1^4 + 6x_1^2 + \frac{1}{2}x_1 - \frac{1}{2}x_2^2$$

Here, the wells are symmetrically distributed across a grid in the 32-dimensional space, where each pair of dimensions corresponds to a well, and $\mu_i$ is uniformly distributed over the space. The total log probability is proportional to the sum of energies from all wells: $\log p(x) \propto E(x) = \sum_{i=1}^{n_{\text{wells}}} E_{\text{DW}}(x_i)$.

**Double Well 4.** The energy function for the DW-4 dataset was introduced in Köhler et al. (2020) and corresponds to a system of 4 particles in a 2-dimensional space. The system is governed by a double-well potential based on the pairwise distances of the particles. For a system of 4 particles, $x = \{x_1, \ldots, x_4\}$, the energy is given by:

$$E(x) = \frac{1}{2\tau} \sum_{i,j} \left[ a(d_{ij} - d_0) + b(d_{ij} - d_0)^2 + c(d_{ij} - d_0)^4 \right]$$

where $d_{ij} = \|x_i - x_j\|_2$ is the Euclidean distance between particles $i$ and $j$. Following previous work, we set $a = 0$, $b = -4$, $c = 0.9$, and the temperature parameter $\tau = 1$. To evaluate the efficacy of our samples, we use a validation and test set from the MCMC samples in Klein et al. (2024) as the ground truth samples following the practice of previous works (Sadegh et al., 2024).

**Lennard-Jones 13 (LJ-13).** This energy function describes a system of $N = 13$ particles. The configuration of these particles is denoted by $x = \{x_1, \ldots, x_{13}\}$, where each $x_k \in \mathbb{R}^{d_s}$ represents the coordinates of the $k$-th particle in $d_s$-dimensional space ($d_s$ being the number of spatial dimensions per particle). The total potential energy $E(x)$ of the system, from which the log probability $\log p(x) \propto -E(x)$ is derived, comprises pairwise Lennard-Jones interactions (Sadegh et al., 2024) and a harmonic confining term, following the practice of (Klein et al., 2024; Sadegh et al., 2024).

The interaction potential $V(d)$ between any two particles $i$ and $j$ separated by a Euclidean distance $d = \|x_i - x_j\|_2$ is given by the Lennard-Jones potential:

$$V(d) = \epsilon \left[ \left(\frac{\sigma}{d}\right)^{12} - 2\left(\frac{\sigma}{d}\right)^6 \right]$$

Here, $\epsilon$ is the depth of the potential well, and $\sigma$ is the finite distance at which the inter-particle potential reaches its minimum ($-\epsilon$). Following previous works, both $\epsilon$ and $\sigma$ are set to 1. It is important to note that a smoothing strategy was employed to prevent the energy from becoming excessively large at short inter-particle distances during training only. At inference time, we employed the vanilla Lennard-Jones energy without smoothing to ensure fair comparison across benchmark methods. Specifically, for distances $d < r_{\text{min}}$ (where $r_{\text{min}} = 0.8$), we applied a quadratic smoothing interpolation. This approach mitigates the characteristic explosion of energy at short distances observed in this class of energy functions. Similar practices are found in existing works; for instance, (OuYang et al., 2024) and (Sadegh et al., 2024) also implement smoothing or apply a hard cut-off to the energy score to keep it within a defined bound.

The total Lennard-Jones energy for the system of $N = 13$ particles is the sum of all pairwise potentials:

$$E_{\text{LJ}}(x) = \sum_{1 \leq i < j \leq 13} V(\|x_i - x_j\|_2)$$

In addition to the pairwise LJ interactions, the particles are subject to a harmonic confining potential, $E_{\text{harmonic}}(x)$:

$$E_{\text{harmonic}}(x) = \frac{1}{2} \sum_{k=1}^{13} \|x_k - x_{\text{COM}}\|_2^2$$

where $x_{\text{COM}} = \frac{1}{13} \sum_{k=1}^{13} x_k$ is the center of mass of the system.

The total potential energy of the system is thus:

$$E(x) = E_{\text{LJ}}(x) + E_{\text{harmonic}}(x)$$

### E.2 METRICS

We evaluate the methods using the Wasserstein-2 ($\mathcal{W}_2$) distance and the Total Variation (TV), both computed with 1,000 ground truth and generated samples. To compute TV, the support is divided into 200 bins along each dimension, and the empirical distribution over 1,000 samples is used. For GMM-40, we report the metrics $\mathcal{W}_2$ on energy space $\mathcal{E}$ and TV on the data space $\mathcal{X}$. For MW-32, we find that $\mathcal{E} - \mathcal{W}_2$ is unstable and thus report $\mathcal{E}$-TV instead. Given the 32-dimensional nature of MW-32, computing TV is impractical; therefore, we report the $\mathcal{W}_2$ metric on the data space rather than TV. For the $n$-body system DW-4, we do not report any metrics in the data space due to its equivariance. Instead, we assess performance using metrics in the energy space ($\mathcal{E}$-$\mathcal{W}_2$ and $\mathcal{E}$-TV) and the interatomic coordinates $\mathcal{D}$ ($\mathcal{D}$-TV) to account for invariance.

### E.3 TRAINING DETAILS

**Gaussian Mixture Model (GMM-40).** We evaluate our method on a 40-mode Gaussian mixture in $\mathbb{R}^2$ to test multi-modal exploration. The velocity field is parameterized by a 4-layer MLP (256-dimensional hidden layers, Layer Norm, and Swish activations) trained using velocity-guided sequential Monte Carlo with Hamiltonian kernels (3 HMC steps, 5 leapfrog steps, step size $\eta = 0.1$). Initial particles are sampled from $\mathcal{N}(\mathbf{0}, 25\mathbf{I})$, optimized via AdamW ($\beta_1 = 0.9$, $\beta_2 = 0.999$) with learning rate $1e^{-4}$, weight decay $1e^{-6}$, and gradient clipping at $\ell_2$-norm 1.0. Training uses 128-particle batches for $10^4$ epochs (500 steps/epoch) with early stopping, converging significantly before the epoch limit.

**Many Well 32 (MW-32).** We assess scalability in high dimensions using a $2^{32}$-mode Many Well potential on $\mathbb{R}^{32}$, exhibiting exponential mode growth with dimension. The velocity field employs a 4-layer MLP (128-dimensional hidden layers, Layer Norm, and Swish activations) trained via velocity-guided SMC with enhanced Hamiltonian kernels (6 HMC steps, 10 leapfrog steps, step size $\eta = 0.1$). Initialized from $\mathcal{N}(\mathbf{0}, 2\mathbf{I})$, optimization uses AdamW ($\beta_1 = 0.9$, $\beta_2 = 0.999$) with learning rate $1e^{-4}$, weight decay $1e^{-6}$, and $\ell_2$-gradient clipping at 1.0. Training maintains 128-particle batches across $10^4$ epochs (500 steps/epoch) with early stopping.

**Double Well 4 (DW-4).** We assess performance in a particle-like system using a DW-4 potential on Euclidean space. The velocity field employs a 4-layer MLP (512-dimensional hidden layers, Layer Norm, and Swish activations) trained via velocity-guided SMC with enhanced Hamiltonian kernels (10 HMC steps, 10 leapfrog steps, step size $\eta = 0.01$). Initialized from $\mathcal{N}(\mathbf{0}, 2\mathbf{I})$, optimization uses AdamW ($\beta_1 = 0.9$, $\beta_2 = 0.999$) with learning rate $10^{-4}$, weight decay $10^{-6}$, and $\ell_2$-gradient clipping at 1.0. Training maintains 128-particle batches across $10^4$ epochs (500 steps/epoch) with early stopping.

**Lennard-Jones 13 (LJ-13).** We assess performance in a particle-like system using a Lennard-Jones 13 (LJ-13) potential on Euclidean space. The velocity field employs a 6-layer DiT architecture (128-dimensional hidden layers, Layer Norm, and Swish activations) trained via velocity-guided SMC with enhanced Hamiltonian kernels (10 HMC steps, 10 leapfrog steps, step size $\eta = 0.01$). Initialized from $\mathcal{N}(\mathbf{0}, 2\mathbf{I})$, optimization uses AdamW ($\beta_1 = 0.9$, $\beta_2 = 0.999$) with learning rate

$10^{-4}$, weight decay $10^{-6}$, and $\ell_2$-gradient clipping at 1.0. Training maintains 128-particle batches across $10^4$ epochs (500 steps/epoch) with early stopping.

### E.4 BASELINE METHODS DETAILS

**Flow Annealed Importance Sampling Bootstrap (FAB).** We use the official PyTorch implementations for GMM-40 and MW-32 ((https://github.com/lollcat/fab-torch), and the JAX implementations for DW-4 and LJ-13 (https://github.com/lollcat/se3-augmented-coupling-flows). All hyperparameters follow their default settings.

**Iterated Denoising Energy Matching (iDEM).** We use the official PyTorch implementations for all experiments (https://github.com/jarridrb/DEM). Note that iDEM did not evaluate on MW-32 in their original paper. For fair comparisons, we use the same 4-layer MLP to parameterise the score network and retain most hyperparameters from GMM-40 unless modifications are necessary.

**Liouville Flow Importance Sampling (LFIS).** We use the official PyTorch implementations (https://github.com/lanl/LFIS). Following FAB, we use MLPs to parameterise the velocity for synthetic datasets and an EGNNs for n-body systems. We adopt the same initial distribution as NFS$^2$ and keep most hyperparameters from the repository unchanged, unless adjustments are required.

**Learning Interpolations between Boltzmann Densities.** For this method, we employ the exact same network architectures as our models for all tasks. As this is also a PINN-based method with an additional learnable interpolation, we maintain the exact same training hyperparameters for all tasks. That is, the only difference is the introduction of additional network capacity on top of the existing one to learn the path interpolation and $\partial_t \log Z_t$, as specified in the original paper (Máté & Fleuret, 2023). This ensures a fair comparison.

**PINN.** For this method, we trained a velocity field using the PINN objective specified in (Albergo & Vanden-Eijnden, 2025). While the method they proposed is stochastic, for a fairer comparison with our approaches, we utilise the fact that the learned velocity field can be used to deterministically generate samples by integrating the Ordinary Differential Equation (ODE), similar to our methods. Therefore, we named it the PINN method instead of NETS (Albergo & Vanden-Eijnden, 2025), as originally used in the paper, to note this difference in benchmarking methodology. All network and training hyperparameters are identical to our setup, with the exception of the extra network capacity used to learn $\partial_t \log Z_t$.

### E.5 ADDITIONAL TRAINING DETAILS

The training and inference pipelines for this project were implemented leveraging the JAX library (Bradbury et al., 2018) for high-performance numerical computation and automatic differentiation. Neural network models were constructed and managed using Equinox (Kidger & Garcia, 2021), a library designed for building and training neural networks in JAX. Blackjax (Cabezas et al., 2024) were used to implement various MCMC-related methods. For the numerical solution of differential equations, which is crucial for our methods, we employed Diffrax (Kidger, 2021). In addition to these core libraries, specific software versions and open-source code utilised throughout this work are detailed in the Jupyter notebook provided within the supplementary information.

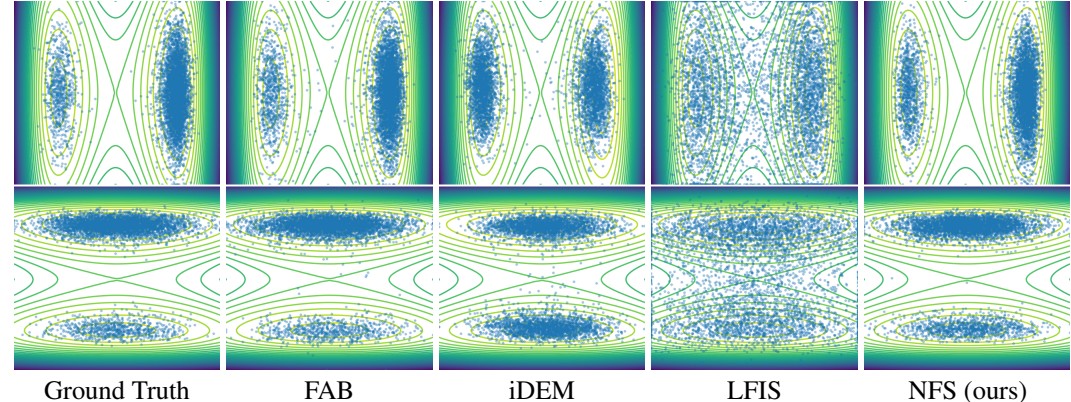

| Ground Truth | FAB | iDEM | LFIS | NFS (ours) |

Figure 12: Samples on MW-32. First row: 2D marginal samples from the 1st and 4th dimensions; Second row: 2D marginal samples from the 2rd and 3rd dimensions.

## F    ADDITIONAL EXPERIMENTAL RESULTS

### F.1    VISUALISATION OF MW-32

This section presents additional visualizations of generated samples on MW-32. As shown in Figure 12, only FAB and NFS$^2$ accurately capture the modes. While iDEM locates the modes, it struggles to identify their correct weights. Additionally, LFIS, another flow-based sampler similar to NFS$^2$, produces noisy samples, highlighting the high variance issue associated with importance sampling. We further illustrate the histogram of sample energy on MW-32, where we draw the empirical energy distribution using 5,000 samples. It shows that NFS$^2$ achieves competitive performance with FAB, and notably outperforms iDEM and LFIS.

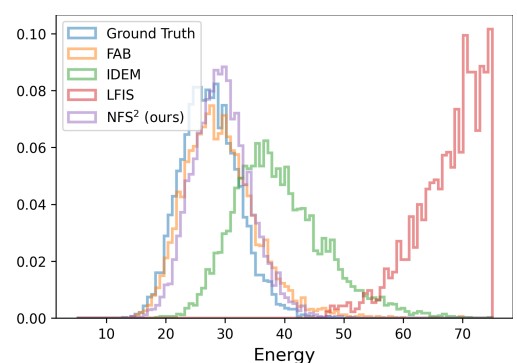

Figure 13: Histogram of sample energy on MW-32.

### F.2    COMPARISONS WITH DIFFERENT SAMPLING STEPS

One key advantage of NFS$^2$ is its ability to achieve high-quality results with fewer sampling steps. In this section, we compare NFS$^2$ to the SOTA diffusion-based sampler iDEM and the flow-based sampler LFIS, using varying numbers of sampling steps. As demonstrated in Figures 15 and 16, NFS$^2$ produces better samples compared to both iDEM and LFIS when using fewer sampling steps.

Moreover, we trained our models using different shortcut regularisation coefficients $\lambda$. As illustrated in Figure 17, the same model trained with stronger shortcut regularisation exhibits stronger performance when the sampling steps are reduced. This is particularly notable for $\lambda = 10$, where even samples generated using only two steps of Euler integration successfully capture all 40 modes of the distribution, whereas the same model trained with much weaker regularisation failed catastrophically. This demonstrates the effectiveness of this approach and paves the way for future research on few-step or even one-step sampling.

### F.3    ANALYSIS OF SAMPLING STEPS ON LJ13

To further investigate the impact of the number of sampling steps on the quality of generated samples for complex, multi-particle systems, we conducted an ablation study on the 13-particle Lennard-Jones (LJ13) cluster. We compare configurations of NFS$^2$, denoted as NFS$^2$-$N_s$ where $N_s$ is the number of

sampling steps, varying $N_s$ from 4 to 128 (using the same trained model). The results are compared against ground truth samples obtained from extensive simulations.

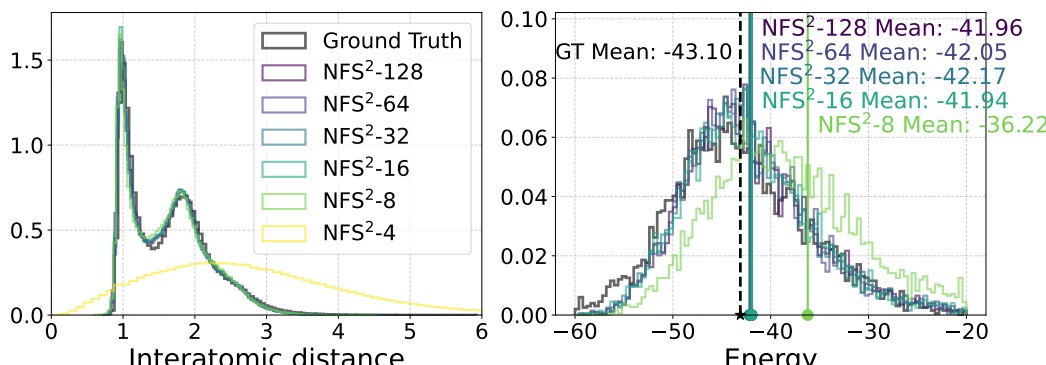

Figure 14: Comparison of (a) interatomic distance distributions and (b) energy distributions for the LJ13 system. Samples are generated by NFS$^2$ with varying numbers of sampling steps (128, 64, 32, 16, 8, and 4) and compared against the ground truth distribution (black). Vertical lines in (b) indicate the mean energies for each method, with markers on the x-axis highlighting these means. The corresponding mean energy values are also annotated.

Figure 14 illustrates the distributions of interatomic distances and energies for the LJ13 system, comparing NFS$^2$ with different numbers of sampling steps against the ground truth.

In panel (a), the interatomic distance distributions show that NFS$^2$ with 128, 64, 32, and 16 steps all closely replicate the multi-modal structure of the ground truth distribution. The NFS$^2$-8 exhibits noticeable deviations, with a lower first peak and a generally broader, less defined structure. The performance degrades significantly for NFS$^2$-4 (yellow), which fails to capture the characteristic peaks and presents a very broad, low-probability distribution shifted towards larger distances.

These results on the LJ13 system demonstrate that while NFS$^2$ can achieve very good approximations of the target distributions with a moderate number of steps (e.g., 16-128), reducing the step count too drastically (e.g., to 8 or 4 steps) can lead to a significant loss in sampling accuracy for such complex energy landscapes. Nevertheless, the performance with 16 or more steps highlights the sampler's efficiency.

## G   LIMITATIONS AND FUTURE WORK

A key challenge of flow-based samplers is the computation of the divergence (see Equation (6)), which becomes prohibitive in high-dimensional settings. While the Hutchinson estimator (Hutchinson, 1989) can be used in practice, it introduces both variance and bias. Alternatively, more advanced architectures can be employed where the divergence is computed analytically (Gerdes et al., 2023). By adopting such architectures, we expect our approach to be scalable to more complex applications, such as molecular simulation (Frenkel & Smit, 2023), Lennard-Jones potential (Klein et al., 2024), and Bayesian inference (Neal, 1993).

Rather than building off of the square error, the objective in Equation (6) can also be formulated using the Bregman divergence, which provides a more general framework for measuring discrepancies and can potentially lead to improved optimization properties, such as better convergence and robustness to outliers. Moreover, while the shortcut model reduces the number of sampling steps required, achieving exact likelihood estimation within this framework remains unclear, presenting a promising direction for future research.

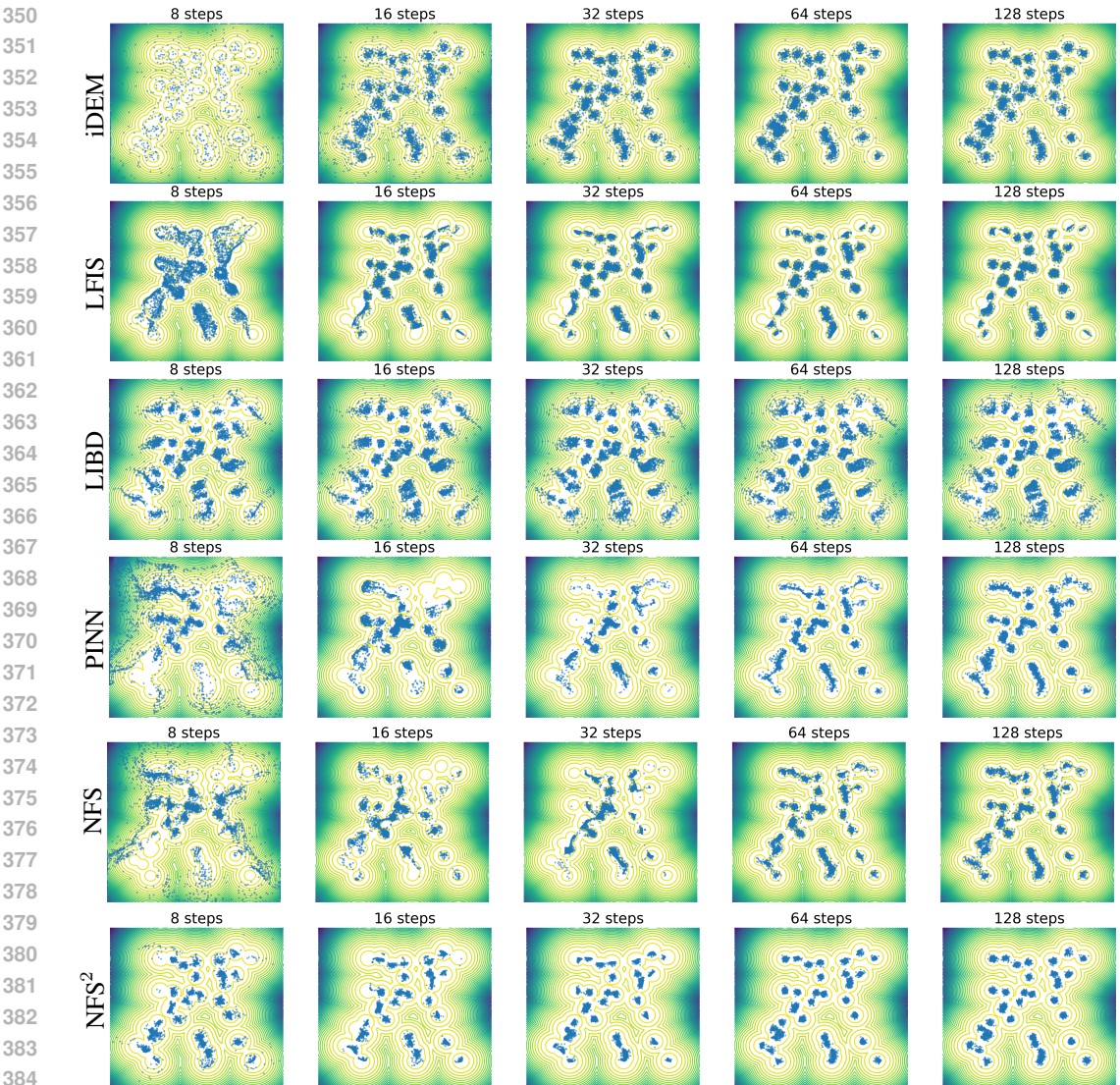

Figure 15: Illustration of the generated samples using different sampling steps on GMM-40.

## H   USE OF LLM

We assert that the use of LLM (Large Language Model) is limited to proofreading and minor stylistic edits. The model did not contribute to the study design, analysis, interpretation, or substantive content in any capacity.

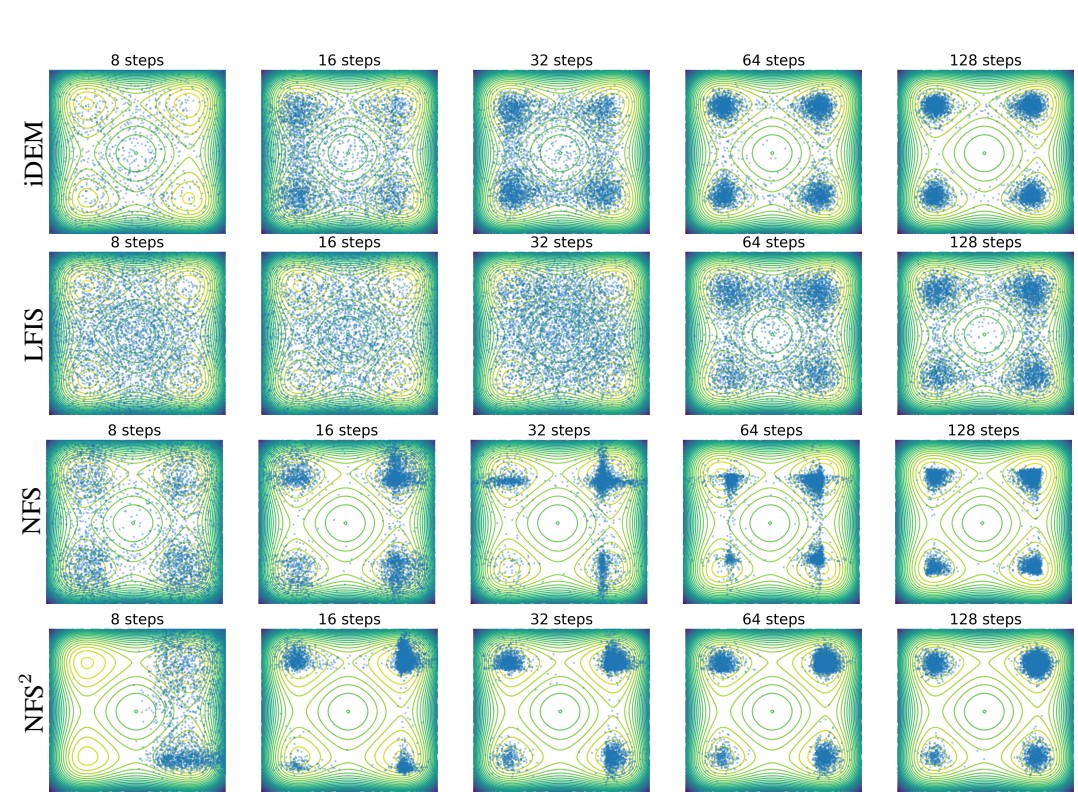

Figure 16: Illustration of the generated samples using different sampling steps on MW-32.

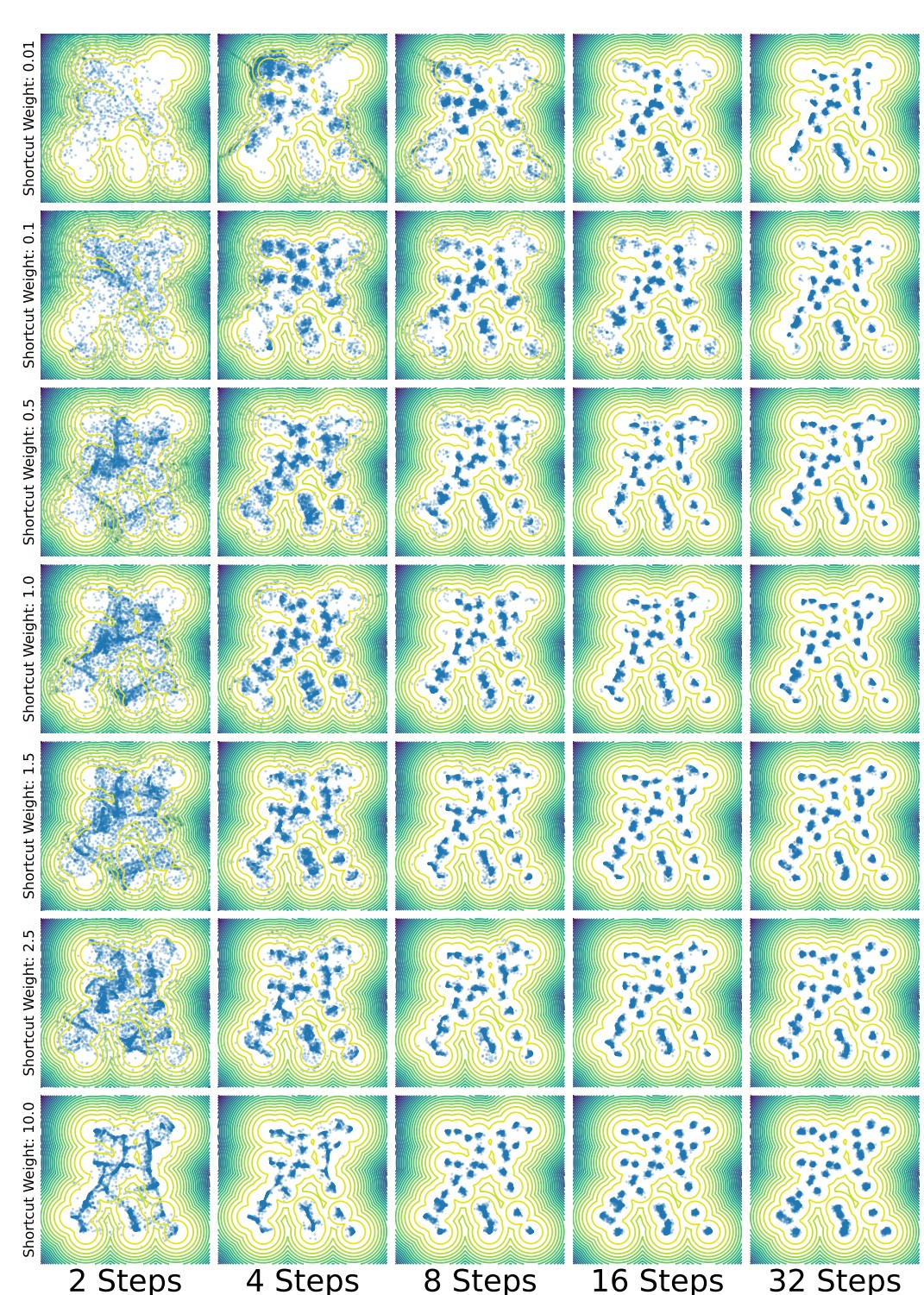

Figure 17: Illustration of the generated samples using models trained with different shortcut regularisation $\lambda$

