# OpenReview forum: "Neural Flow Samplers with Shortcut Models"
_ICLR.cc/2026/Conference — ICLR 2026 Conference Withdrawn Submission_

### Official Review · Reviewer_8EgW · 2025-10-18

**Soundness:** 3
**Presentation:** 3
**Contribution:** 3
**Rating:** 4
**Confidence:** 5

**Summary:**

This work proposes a flow model for sampling from Boltzmann distributions. The core of this work is to train a vector field $v_\theta(x, t)$ to traverse the probability path $(p_t)_{t}$ defined by annealing with a PINN loss induced by the continuity equation. The aid training of such a flow model, the author propose two ingredients.

1. To calculate the PINN loss, it requires the change of log partition function, i.e. its time-derivative $\partial_t\log Z_t$, which is intratable. To estimate it, one usually uses importance sampling, which admits high variance. To alleviate it, the author proposes to use SMC, where the transition kernel in the SMC is informed by the velocity network.

2. To further accelerate the sampling process, the author proposes to parameterize a shortcut model and train it jointly with the shortcut consistency loss.

By combining those two proposed ingredients, the author shows that the proposed method can achieve comparable performance with fewer function evaluations during sampling, compared to other baselines.

However, there're several concerns, which will be stated in the "Weakness" and "Questions" sections.

In summary, the reviewer suggests a weak reject of this paper.

**Strengths:**

1. The paper is clearly written

2. The proposed control variate is shown to be effective, with a lower variance of the estimation.

3. It is a good contribution to explore the few-step generative models in the region of neural sampler, as most methods are more focus on the quality of samples and evaluations of energy functions, rather than the efficiency of the sampling process.

4. Training neural samplers that generate correct weights of modes are important but usually underestimated in many related works. This paper considers the MW32 case which is difficult for score-matching based methods, such as iDEM. The experimental results show that the proposed method could recover the correct proportion, which appears to be more important than the LJ13 experiment to the reviewer.

**Weaknesses:**

1. The first concern is the estimation of $\partial_t\log Z_t$. It is well-known that estimating the log partition function, aka free energy, is a fundamental and challenging problem in statistical sciences. An accurate estimation of the free energy is usually hard and not scalable to high dimensional spaces. Intuitively speaking, if $\partial_t\log Z_t$ is accurate and $\partial_T\log Z_T$ is tractable, then $\log Z_0$ can be estimated through integration with small accumulated bias, which is called Thermodynamic Integration (TI) [1]. However, TI is not quite scalable and therefore the reviewer suspects the same scalability of the proposed estimation.

2. Regarding estimation of $\partial_t\log Z_t$, it requires samples from the marginals $p_t$, which requires to simulate MCMC as illustrated in the paper. That means, by running MCMC for  $\partial_t\log Z_t$ estimation, the author has access to the marginal samples, which raises two sub-questions:
    1. If the author trusts the samples generated by the proposed SMC, that means access to $(x_t)_t$ with $x_t\sim p_t$ is available. And the number of samples from each marginal should be a lot since they are used for an accurate estimation of  $\partial_t\log Z_t$. In such case, one could always train standard generative models.
    2. If not, it is not clear that how this mismatch is corrected

3. The usage of shortcut model, instead of other few-step generative models such as Consistency models [2], Consistency Trajectory Models [3], and the recent Mean-flow model [4], is not illustrated. As it appears straightforward to the reviewer to replace the shortcut loss with other distillation losses. An illustrative paragraph for clarification would be great.

4. Regarding scalability, instead of the accuracy of  $\partial_t\log Z_t$ estimation, another concern lies on the trace calculation of the velocity net, which is computationally expensive. A comparison between the training overheads and practical timing comparison is necessary. On the other hand, the tasks considered in the experiments are relatively simple: for GMM cases, a 50 dimensional one would be more desired as in [4]; for LJ, the 55-particle one is important for showcasing the scalability, especially the author introduces the trace calculation of the velocity net in training.

5. Since the marginals $(p_t)_t$ are predefined by annealing, and the traversal (the velocity net) is trained, one could employ SMC during sampling to debias the mismatch introduced by training error. It would be valuable to explore how this debias scheme could further correct the trained model with a few steps in generation.

6. The number of energy function is not declared in the manuscript, which is important for training neural samplers. Especially, the author proposes to run SMC to estimate $\partial_t\log Z_t$.

7. Some minor issues: figure 5 and 6 are not clear, the author should reorganize the legands and labels to reduce overlap and out-side-of-box.

[1] Máté, Bálint, François Fleuret, and Tristan Bereau. "Solvation free energies from neural thermodynamic integration." The Journal of Chemical Physics 162.12 (2025).

[2] Song, Yang, et al. "Consistency models." (2023).

[3] Kim, Dongjun, et al. "Consistency trajectory models: Learning probability flow ode trajectory of diffusion." arXiv preprint arXiv:2310.02279 (2023).

[4] Chen, Junhua, et al. "Sequential controlled langevin diffusions." arXiv preprint arXiv:2412.07081 (2024).

**Questions:**

see weakness.

---

### Official Review · Reviewer_2jMW · 2025-10-21

**Soundness:** 3
**Presentation:** 3
**Contribution:** 2
**Rating:** 4
**Confidence:** 4

**Summary:**

The paper deals with sampling from an unnormalized density using a kind of continuous normalizing flow trained to follow interpolation paths in the space of densities between an easy base distribution and the target. The proposed method draws inspiration in particular from (Máté & Fleuret, 2023; Tian et al., 2024; Albergo & Vanden-Eijnden, 2025) and seeks to minimize a PINN-like loss derived from the continuity equation. Going beyond these references, it is  proposed to estimate the time derivative of the log partition function using a Sequential Monte Carlo and an appropriate variance reduction. Additionally, the continuous-time model is replaced by a model predicting the velocity field over a given time interval in order to speed up the generation by decreasing the number of integration time steps. The proposed method is compared to the PINN strategies of the references it draws inspiration from (LFIS, LIBD and PINN-NETS) and a sampler using diffusion models (IDEM) and a lot one using normalizing flows and AIS.

**Strengths:**

1. The paper proposes novel improvements for a class recently proposed samplers that rely on ideas coming from generative modelling.
2. An ablation study of the proposed improvements is presented and encouraging although each test is only performed in one experiment on one system.

**Weaknesses:**

3. There is no discussion on the computational budget of the method compared to related methods in general and more specifically in each numerical experiment proposed neither at the stage of training nor at the stage of sampling once training is done. For instance, at least in the main text, there is no discussion of the number of integration steps for other methods. It is therefore difficult to interpret the performance comparison between methods and the benefits of the proposed improvements.
4. There seems to be an inconsistency in the proposed estimation of the (derivative of the) log partition function (see Question 6 below).
5. The paper reproduces a common flaw of the adjacent literature that is to neglect to check the performance of the proposed algorithm in systematic experiments. As the number of papers increases in this line of work, proposing each time different adjustments, demanding yet easily interpretable benchmarks are crucially needed. For instance, reporting accuracy in mode weights on Gaussian mixtures systematically on a collection of experiments where the size and properties of the mixture can be varied as proposed in [Grenioux2025] (and not only by visual inspection of MW32) would here be very relevant. It is a straightforward metric and expected to be particularly challenging for the proposed approach that relies on interpolation paths in densities that can feature a mode switching behavior (Maté & Fleuret 2023). Such simple experiments would greatly strengthen the paper.

Minor:
- Citing in the introduction the works that pioneered using generative models for sampling could be a good idea [Noé2019,Wu2019,Albergo2019].
- At the beginning of Eq (2), it should be a total derivative $d\log p_t(x_t)$ in th LHS, this is just a rewriting of the continuity equation (3-4)
- $\tilde w$ is not defined before being used in the definition of the ESS, it is probably a $\bar{w}$? (Line 185)
- An equal sign appears to be missing on line 218.
- Which task, with which experimental set up, is Figure 2 related to?
- Repetition on line 337 and 374 Table Table
- What do the authors mean by “sorted” interatomic distances? (Line 354)
- Figure 5 and 6 are difficult to read and should be made bigger given that the page count is not reached.

References:
- [Grenioux 2025] Grenioux, Louis, Maxence Noble, and Marylou Gabrié. “Improving the Evaluation of Samplers on Multi-Modal Targets.” Paper presented at Frontiers in Probabilistic Inference: Learning meets Sampling. ICLR Workshop on Frontiers in Probabilistic Inference: Learning Meets Sampling, April 24, 2025. https://openreview.net/forum?id=d91E9RhVFU.
- [Noé 2019] Noé, Frank, Simon Olsson, Jonas Köhler, and Hao Wu. “Boltzmann Generators: Sampling Equilibrium States of Many-Body Systems with Deep Learning.” Science 365, no. 6457 (2019): eaaw1147. https://doi.org/10.1126/science.aaw1147.
- [Wu2019] Wu, Dian, Lei Wang, and Pan Zhang. “Solving Statistical Mechanics Using Variational Autoregressive Networks.” Physical Review Letters 122, no. 8 (2019): 080602. https://doi.org/10.1103/PhysRevLett.122.080602.
- [Albergo 2019] Albergo, M. S., G. Kanwar, and P. E. Shanahan. “Flow-Based Generative Models for Markov Chain Monte Carlo in Lattice Field Theory.” Physical Review D 100, no. 3 (2019): 034515. https://doi.org/10.1103/PhysRevD.100.034515.

**Questions:**

6.  The proposed approach bears similarities with Annealed Flow Transport (AFT) [Arbel2021], and the follow up CRAFT [Matthews2022], which relied on normalizing flows, but that appear relevant to cite as related works. Here the velocity based initialization in SMC plays the role of the learned intermediate transport map in AFT. I might be missing something, but this parallel seems to evidence that the proposed SMC with the velocity-based initialization is not strictly sampling from the target and therefore that the statistical estimation of the partition function should not be unbiased. The missing bit is to take into account this additional deterministic update that adds the determinant of the Jacobian of the transformation to the weight update (Equation 9 from Arbel 2021). It does not seem trivial to compute this determinant in the proposed approach if it is indeed needed. The influence of this inconsistency is maybe minimal, but the authors could check this looking at synthetic examples with known partition functions such as Gaussian mixtures (with increasing dimensions).

7. Figure 1: As the number of MCMC steps per SMC step is increased, the advantage of the learned initialization seems to disappear. Could the authors comment on this fact? Maybe the inconsistency reported above starts showing here?

8. The description of the results on GMM-40 of $\mathcal{E}- W_2$ does not seem to match the numbers in Table 1. The numbers are more in favor of the proposed method than what the text says. Were these results updated last minute and the authors forgot to update the accompanying text? Can they be trusted?

9. For their synthetic experiments, why do the authors report $\mathcal{X}- W_2$, $\mathcal{E}-TV$ in one case and  $\mathcal{E}- W_2$, $\mathcal{X}-TV$  in the other case? Reporting the 4 metrics each time would be more complete.

10. In their proposition, Albergo & Vanden Eijnden 2025 (NETS) also provide a reweighing strategy for the samples generated with the learned velocity field. Have the authors included these in their benchmark or simply the outputs of the continuous normalizing flow?

- [Arbel2021] Arbel, Michael, Alex Matthews, and Arnaud Doucet. “Annealed Flow Transport Monte Carlo.” Proceedings of the 38th International Conference on Machine Learning, PMLR, July 1, 2021, 318–30. https://proceedings.mlr.press/v139/arbel21a.html.
- [Matthews2022] Matthews, Alex, Michael Arbel, Danilo Jimenez Rezende, and Arnaud Doucet. “Continual Repeated Annealed Flow Transport Monte Carlo.” Proceedings of the 39th International Conference on Machine Learning, PMLR, June 28, 2022, 15196–219. https://proceedings.mlr.press/v162/matthews22a.html.

---

### Official Review · Reviewer_3Rm5 · 2025-10-30

**Soundness:** 3
**Presentation:** 1
**Contribution:** 2
**Rating:** 2
**Confidence:** 4

**Summary:**

The paper addresses the problem of sampling from a target density when we only have access to the unnormalized density and no samples from it. The authors propose to adapt shortcut models to this setting by transporting samples between marginals of a linear interpolation of the densities in log-space to enable faster sampling, and by introducing an improved estimator for $\partial_t \log Z_t$ that uses the learned velocity to move samples between marginals and HMC to mix samples within these marginals. The method is evaluated on toy systems (DW-4, LJ13, GMM-40, and MW-32) and includes several ablations on model architectures and regularization for shortcut learning.

**Strengths:**

1. Adapts the shortcut-modeling idea to a flow/SMC-based sampling setup, showing it can be used to reduce the number of transport steps.

2. Achieves consistently strong results on the reported toy and small N-body benchmarks (DW-4, LJ-13, GMM-40, MW-32).

3. Empirically shows that the proposed estimator yields lower-variance estimates of $\partial_t \log Z_t(x)$ during training, which is important for stabilizing PINN-style objectives.

**Weaknesses:**

1. The paper is hard to follow and poorly structured. It reads like a mix of several ideas and lacks a cohesive structure. The title focuses on shortcut models, but the first thing introduced is variance reduction for $\partial_t \log Z_t​$, rather than the shortcut method, which seems to be the main topic according to the title. Reordering the sections would help where the method should be introduced first and then talk about improving stability by having a lower variance estimator of $\partial_t \log Z_t(x)$ as an additional benefit.

2. While the paper shows strong empirical performance on the toy targets, the theoretical contribution is comparatively modest, since it mainly adapts shortcut modeling to this particular SMC/flow-based sampling setup rather than introducing something fundamentally new. That said, showing that shortcut models can be made to work in this setting is still interesting

3. he experiments are restricted to synthetic or small N-body systems (DW-4, LJ-13, GMM-40, MW-32), so it is hard to assess whether the method remains effective on practical sampling problems such as Boltzmann sampling of peptide conformations, either in dihedral space as in [1] or in Cartesian space as in [2]. A demonstration on at least one peptide-like system (e.g., alanine dipeptide with torsions, or a small capped peptide in Cartesian coordinates) would make the empirical claims more convincing.

4. Below are comments about the presentation of results and figures:

    a. Figure 2: add spacing between the caption and the following text.

    b. Figures 5 and 6: currently poorly visualized, I would suggest removing the bars with the means; Table 1 should highlight better performance (bolding or coloring).

    c. Table 1: metrics are inconsistent (e.g., E-TV missing for GMM-40, E-W2 missing for MW-32). It would be better to make the metrics uniform across tasks and to bold best results.

    d. The architecture ablations are better placed in the appendix, since the method is architecture-agnostic and mainly benefits from using the best-performing architecture (which appears to be a transformer). The main text should focus more on shortcut ablations and on the importance of accurate $\partial_t \log Z_t$ estimates.

    e. The results are missing ESS metrics.


[1] Midgley, Laurence Illing, et al. "Flow annealed importance sampling bootstrap." arXiv preprint arXiv:2208.01893 (2022).

[2] Akhound-Sadegh, Tara, et al. "Progressive Inference-Time Annealing of Diffusion Models for Sampling from Boltzmann Densities." arXiv preprint arXiv:2506.16471 (2025).

**Questions:**

1. In Figure 1, what exactly does “SMC” denote? Is this a sequential Monte Carlo pipeline that uses $\nabla \log p_t(x)$ inside HMC to move particles between intermediate marginals, rather than using the learned velocity field?


2. Do the authors have an explanation for why “Velocity + SMC” underperforms at 10 steps in Figure 1?

3. It would help to include an ablation comparing (a) Velocity + SMC + learned $\partial_t \log Z_t$ vs. (b) Velocity + SMC + Stein control variates. Right now the paper shows that the Stein-based estimator reduces variance during training, but it is not clear how much of that translates to improved inference-time performance compared to simply learning $\partial_t \log Z_t$. This would also clarify the difference from NETS, which I believe those authors estimate $\partial_t \log Z_t$ via the reweighted expectation of $\nabla \cdot v_t(x) + v_t(x) \cdot \nabla \log p_t(x) + \partial_t \log \hat{p}_t(x)$, rather than using a separate trained predictor [3].


4. Could the authors describe in more detail the augmentation strategy that samples proportional to the residual error, and provide an ablation on how this affects training stability? It would be useful to have the exact procedure (either as equations or short pseudocode) in the appendix.


5. How does the method perform without resampling during the transport? This would help isolate the contribution of the learned velocity $+$ HMC move from the resampling step.


6. What are the ESS values for the proposed model (and for the baselines)?


7. Have the authors tested the method on a larger Lennard–Jones system such as LJ-55? This would illustrate how well the approach scales beyond the toy/N-body setups shown.


**Typos**

1. Lines 218–219: there seems to be a missing “=” in the expression involving $\partial_t \log Z_t , \mathbb{E}[\cdot]$.

[3] Albergo, Michael S., and Eric Vanden-Eijnden. "Nets: A non-equilibrium transport sampler." arXiv preprint arXiv:2410.02711 (2024).

---

### Official Review · Reviewer_t1NB · 2025-11-03

**Soundness:** 3
**Presentation:** 3
**Contribution:** 2
**Rating:** 4
**Confidence:** 4

**Summary:**

The approach introduces a new velocity-based SMC method designed to reduce sampler variance when drawing samples from unnormalized probability densities. NFS$^2$ also integrates self-consistency terms into the loss function to alleviate mode collapse in few-step sampling settings, enabling efficient sampling with lower inference costs compared to conventional diffusion- and flow-matching-based methods.

**Strengths:**

The method is principled in nature, and the few-step consistency approach is practically useful for improving sampler inference efficiency, especially when attempting to scale up to larger systems.

**Weaknesses:**

The performance of the method on energy-W2 appears quite variable across all tested potentials, particularly in the few-step regime (8 steps). To strengthen the evaluation, the method could be benchmarked on a larger and more complex system like LJ55, which is commonly used for comparison in prior works (e.g., iDEM and other methods use this as a baseline). It would also be valuable to compare the approach against newer state-of-the-art methods such as adjoint sampling, which has recently been shown to outperform iDEM. The speed advantage of the proposed method remains unclear, as no quantitative comparison has been demonstrated. Additionally, including other performance metrics such as ESS would provide a more comprehensive assessment of the method’s efficiency and accuracy.

**Questions:**

- Can other one-step or few-step sampling strategies (e.g., MeanFlow, AYF, etc.) be incorporated into the proposed approach to further improve sampling performance? These generalized versions of shortcut models might outperform the simpler shortcut variants. It would be interesting to see an ablation contrasting these approaches.
- For the diffusion- and flow-matching-based solvers compared against (iDEM etc.), how many steps are used to integrate the velocity fields? Some comparisons involve showing 128-step NFS, which may end up using a similar number of steps?
- Could the ESS also be reported for a more comprehensive assessment?
- How does the approach perform relative to adjoint sampling (arXiv preprint arXiv:2504.11713)---an approach seen to outperform iDEM?
- The method benchmarking is good, but including results on a more complex potential would strengthen the analysis. Experiments on LJ55 and perhaps a simple peptide system such as alanine dipeptide could better demonstrate scalability and robustness.
- It would be helpful to ascertain how much faster the few-step samplers are compared to the other baselines. Would it be possible to include an ablation study quantifying the speedup achieved by the few-step samplers relative to the other baseline methods?

---

### Note · Authors · 2025-11-28

**Comment:**

Dear Reviewers,

We have decided to withdraw this manuscript in order to substantially improve the work, incorporating the valuable feedback and suggestions provided in your reviews. We are sincerely grateful for the time and care you devoted to evaluating our submission, and we hope to come back with a much stronger version in the future.

Many thanks.

**Withdrawal Confirmation:**

I have read and agree with the venue's withdrawal policy on behalf of myself and my co-authors.